# SOTOPIA-RL: REWARD DESIGN FOR SOCIAL INTELLIGENCE

## ABSTRACT

Social intelligence has become a critical capability for large language models (LLMs), enabling them to engage effectively in real-world social tasks such as collaboration and negotiation. Reinforcement learning (RL) is a natural fit for training socially intelligent agents because it allows models to learn sophisticated strategies directly through social interactions without requiring human annotations. However, there are two unique parts about social intelligence tasks: (1) the quality of individual utterances in social interactions is not strictly related to final success; (2) social interactions require multi-dimensional rubrics for success. Therefore, we argue that it is necessary to design rewards for building utterance-level multi-dimensional reward models to facilitate RL training for social intelligence tasks. To address these challenges, we propose SOTOPIA-RL, a novel framework that refines coarse episode-level feedback into utterance-level, multi-dimensional rewards. Utterance-level credit assignment attributes outcomes to individual utterances, while multi-dimensional rewards capture the full richness of social interactions and reduce reward hacking. Experiments in SOTOPIA, an open-ended social learning environment, demonstrate that SOTOPIA-RL achieves state-of-the-art social goal completion scores (7.17 on SOTOPIA-hard and 8.31 on SOTOPIA-full), significantly outperforming existing approaches. Ablation studies confirm the necessity of both utterance-level credit assignment and multi-dimensional reward design for RL training.

## 1 INTRODUCTION

Social intelligence (Gweon et al., 2023; Mathur et al., 2024; Zhu et al., 2025), a capability with applications in customer service (Pandya & Holia, 2023; Bamberger et al., 2023), educational tutoring (Stamper et al., 2024; Nye et al., 2023), conflict resolution (Aggrawal & Magana, 2024), and team coordination (Li et al., 2023; Guo et al., 2024), has emerged as a crucial capability for large language models (LLMs). Social intelligence is naturally formed via social interactions, and Reinforcement learning (RL) is a natural fit for training socially intelligent agents because it allows models to learn sophisticated strategies directly through social interactions. Therefore, utilizing reinforcement learning (RL) for social intelligence learning is natural. Although prior work has shown that RLs can be used to optimize LLM abilities for math (Shao et al., 2024b; Yang et al., 2024a), coding (Hui et al., 2024; Wei et al., 2025), and reasoning (Guo et al., 2025), tuning mainstream LLMs (Yang et al., 2025; Touvron et al., 2023; Hurst et al., 2024) for social intelligence with RL remain underexplored (Zhou et al., 2025; Mathur et al., 2024). We argue that a central obstacle is the lack of a unified and effective reward design that can be tailored to social behavior and be applied to diverse types of social scenarios like collaboration, negotiation and accommodation. In this paper, we propose a practical reward-design recipe for social intelligence and apply reinforcement learning to train social agents that generate high-quality social utterances.

**Uniqueness of social intelligence tasks**. Unlike math and coding tasks, which often provide clear and verifiable rewards (Shao et al., 2024b; Wei et al., 2025; Cui et al., 2025), social tasks (*e.g.*, persuasion and collaboration) present fundamentally different challenges for reward design. First, the quality of individual utterances in social interactions is hard to define and often *only loosely correlates* with the final success: in a negotiation, for example, a social agent might deploy a misleading claim that nevertheless helps secure a better outcome. By contrast, domains such as math or coding give much clearer signals, where correct intermediate steps are usually required for a correct out-

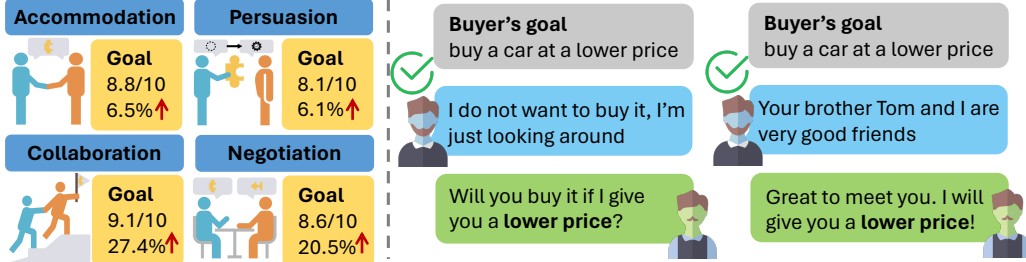

Figure 1: **Uniqueness of social intelligence tasks. (Left)** Accommodation, persuasion, collaboration, and negotiation represent four core types of social intelligence tasks. Our method achieves consistent improvements across all tasks compared with SOTOPIA-π, the previous state-of-the-art on SOTOPIA. **(Right)** Two unique features of social interactions: (1) utterances are not always directly tied to outcomes—*e.g.*, people may lie or mislead in negotiations; (2) interactions are inherently multi-dimensional—*e.g.*, relationship building can play a crucial role in achieving task success.

come. Second, social interactions are *inherently multi-dimensional*: some utterances in the social interactions directly advance the task goal, while others play an indirect but crucial role by building rapport, maintaining engagement, or preserving conversational flow. Unlike largely verifiable, often binary outcomes in math or coding (Su et al., 2025), social interactions must be considered and evaluated across multiple interacting dimensions.

**Necessity of utterance-level multi-dimensional reward models for social intelligence tasks.** Training social agents with only episode outcomes is common in principle but is highly sample-inefficient and often hard to capture fine-grained conversational behaviors. It is mainly because utterance-level contributions weakly correlate with episode outcomes. Instead, we argue for utterance-level, multi-dimensional reward models (RM): first, episode outcomes can be decomposed into utterance-level contributions, and LLMs are capable of reliably inferring which utterances helped or hindered success—empirically, diverse LLM families show strong agreement in attribution (with Spearman correlations > 0.7). Second, evaluating utterance quality along separate sub-dimensions (*e.g.*, goal achieving, relationship building, knowledge-sharing) builds a rubric-based assessment for difficult holistic judgement, which simplifies credit assignment and reduces variance in reward estimates. Together, these observations explain why multi-dimensional, utterance-level reward design is both practical—often without costly human annotation—and better aligned with human preferences than monolithic episode-level ones (Ram'e et al., 2024; Moskovitz et al., 2023).

**Our method.** In this paper, we propose SOTOPIA-RL, a novel and practical RL training receipt for social agent. It includes two main stages: (1) offline social reward collection; (2) online social agent training. For the first stage, we collect a set of existing social interaction episodes with outcomes and utilize LLMs and multi-dimensional rubrics to conduct utterance-level, multi-dimensional social reward assignment. For the second stage, we utilize such offline-collected social reward to train an utterance-level, multi-dimensional RM and conduct online RL training.

**Main discoveries.** To prove the effectiveness of our proposed RL methods, we evaluate based on SOTOPIA (Zhou et al., 2023a), an open-ended social learning environment. SOTOPIA provides diverse social tasks and multi-dimensional social evaluation. Experiments conducted in the SOTOPIA environment reveal two key findings: (1) Social agents trained with SOTOPIA-RL consistently outperform all baselines on social goal completion metrics provided by SOTOPIA, achieving a goal completion score of 7.17 on the SOTOPIA-hard benchmark and 8.31 on the full SOTOPIA dataset. (2) Our utterance-level and multi-dimensional reward design is critical for stable and effective RL training in complex social scenarios. These results highlight the importance of social reward design and validate the core design principles behind SOTOPIA —particularly the importance of evaluating social interaction quality across diverse dimensions.

## 2 RELATED WORK

**Utterance in social interactions.** A social utterance is a deliberate communicative act produced to pursue social goals (Bahing et al., 2018). Social interactions are sequences of such utterances, with

multi-turn exchanges forming an episode. Beyond linguistic tokens, utterances convey intentions and emotions that shape the conversation (Ghosal et al., 2020; Wang et al., 2022).

**Social skill learning**. To enhance agents' social intelligence, prior approaches have leveraged RL in different ways. SOTOPIA-$\pi$ (Wang et al., 2024b) adopts self-reinforcement learning, Ndousse et al. (2021) use conversation-level rewards, and Stable Alignment (Pang et al., 2024) relies on rule-based peer feedback without explicit rewards. SDPO (Kong et al., 2025) incorporates preference-based tuning but ignores utterance-level effects. Our contribution is to introduce utterance-level reward modeling tailored for social tasks. Rather than injecting explicit strategies during training (Zhang et al., 2025; Wang et al., 2025), we capture social skills implicitly through the reward design.

**Process reward modeling**. For RL with verifiable rewards (RLVR) such as math and programming, Process Reward Models (PRMs) have proven effective. PRIME (Cui et al., 2025) assigns token-level rewards from outcome labels, boosting reasoning without explicit process annotations. Other works (Choudhury, 2025; Wang et al., 2024a) use Monte Carlo (MC) rollouts to compute reward targets, where repeated sampling estimates expected values in stochastic or complex decision-making tasks (Barto, 2021). Designing utterance-level rewards for social tasks can be viewed as a form of process reward, but unlike math or programming, the ambiguity and multi-dimensionality of social interactions demand multi-dimensional evaluation.

**Multi-objective RL**. Multi-objective RL aligns LLMs with multiple preferences by optimizing over several reward functions. Most approaches use linear scalarization to combine rewards into a single objective (Jang et al., 2023; Yang et al., 2024b; Li et al., 2020; Zhou et al., 2023b), while recent work explores non-linear utilities (Cheng et al., 2025; Xie et al., 2024) and reward decomposition (Mao et al., 2025; Shenfeld et al., 2025; Lee et al., 2024). Building on these directions, we focus on multi-dimensional reward learning for social tasks, using linear scalarization where auxiliary rewards (*e.g.*, relationship maintenance, knowledge seeking) explicitly support goal completion.

# 3 SOCIAL LEARNING ENVIRONMENT

To enable RL training, we first need to define a suitable environment. Such an RL environment must serve two purposes: (i) it should **generate data** by simulating social interactions, and (ii) it should **provide feedback** that can be used as training signals. Following SOTOPIA (Zhou et al., 2023a), we implement both aspects: interaction trajectories are generated through multi-turn dialogues between agents, and episode-level feedback is provided through LLM-based evaluation.

## 3.1 SOCIAL INTERACTION

Social interactions are dialogues between a pair of agents. From the perspective of a single agent, another side of the social agent is considered as the *partner model*. The interaction process can be described as a *partially observable Markov decision process* (POMDP), represented by the tuple $\langle \mathcal{S}, \mathcal{A}, \mathcal{O}, T, Z, R \rangle$. $\mathcal{S}$ represents the set of possible social states, $\mathcal{A}$ represents the action space, $\mathcal{O}$ represents the observation space. $T : \mathcal{S} \times \mathcal{A} \to \mathcal{S}$ is the transition function that captures how the social state evolves given the agent's utterance and the partner's response. $Z : \mathcal{S} \to \mathcal{O}$ is the observation function. $R : \mathcal{S} \times \mathcal{A} \to \mathbb{R}$ is the reward function that reflects the overall quality of the social interaction with respect to the agent's private goal, such as successful persuasion or mutual understanding.

**Observation space**. In a social learning environment like SOTOPIA, the observations refer to the history of dialogue. The social agent operates under partial observability, receiving at each time step $t$ a private observation $o_t \in \mathcal{O}$. $\mathcal{O}$ consists of all dialogue histories and contextual cues that the agent can perceive, but excludes latent variables such as the partner's private goals, beliefs, or emotions. The observation $o_t$ is generated by the probabilistic observation function $Z$, modeling the partial and asymmetric nature of social perception. A social episode with $T$ turns is defined as

$$\tau = \big(o_0, a_0, o_1, a_1, \ldots, o_T\big), \tag{1}$$

where $o_t \in \mathcal{O}$ is the dialogue history observed at time $t$, and $a_t \in \mathcal{A}$ is the utterance generated by the agent at time $t$. This episode captures the full sequence of observations and actions.

**Action space**. An action in SOTOPIA is defined as an utterance, including both verbal (*e.g.*, speaking) and non-verbal (*e.g.*, smiling, hugging) communication. The action space $\mathcal{A}$ contains all such communicative behaviors available to the agent. At turn $t$, the agent samples $a_t \sim \pi_\theta(\cdot \mid o_t, g)$

from its policy conditioned on the current observation $o_t$ and its private goal $g$. The chosen action, together with the partner's response, contributes to and becomes part of the next observation $o_{t+1}$.

**MDP approximation**. To optimize the social agent with MDP-based RL methods like GRPO (Shao et al., 2024b), we adopt an MDP approximation. At step $t$, the state $s_t$ is considered as the dialogue history together with the agent's private social goal. The social policy $\pi_\theta$ outputs a distribution over utterances $a_t$ (*i.e.*, $a_t \sim \pi_\theta(\cdot \mid s_t)$), and offline rewards $r_t$ are used to train an online utterance-level reward model $R_\psi(s_t, a_t)$.

### 3.2 SOCIAL EVALUATION

As a social learning environment for RL training, it should be able to provide quantitative feedback on the outcome of the social interaction. Therefore, we adopt the multi-dimensional social evaluation in SOTOPIA as environment feedback. In particular, we define the goal completion score $G$ as the primary training signal, obtained from an LLM-based evaluator $f_\theta$ conditioned on the social episode $\tau$ and the agent's private goal $g$:

$$G = f_\theta(\tau, g) \in \mathbb{R}. \qquad (2)$$

Beyond goal completion, it also produces six auxiliary evaluation dimensions—believability, knowledge seeking, relationship maintaining, secret keeping, social rule-following, and financial/material benefits—following the SO-TOPIA rubric. These additional dimensions are carefully defined for analysis and provide richer insights into the quality of social interactions. Detailed definitions for each dimension are given in Appendix §I.

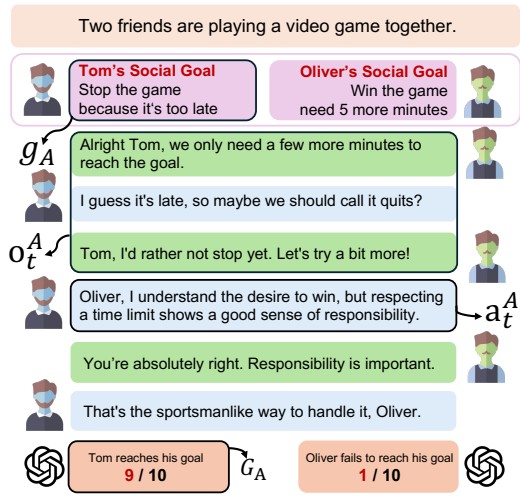

Figure 2: **An example of a social task in the SOTOPIA environment**. Tom is agent A, and Oliver is agent B. Each agent has a unique goal that is hidden from the other. "9 / 10" indicates a single-dimensional episode-level reward provided by LLMs to describe its goal completion status.

## 4 METHODOLOGY

In Figure 3, we introduce the proposed SOTOPIA-RL framework, which consists of two stages: (1) offline social reward design with LLMs and (2) online social agent training with RL. In the offline stage, high-quality reward labels are generated using the entire dialogue episode, where each utterance is annotated by LLMs with access to both its preceding and subsequent context. These offline labels are then distilled into an online reward model (RM) that relies only on the dialogue history available up to the current utterance. In the online stage, this RM is used to provide real-time reward signals during interaction, enabling policy optimization through online RL.

### 4.1 SOCIAL REWARD DESIGN WITH LLMS

Providing accurate utterance-level rewards for social interactions is challenging due to the uniqueness of social intelligence tasks mentioned in Section §1. We address this challenge in two steps, as shown in Figure 4. First, through **reward attribution**, we assign episode-level outcomes to individual utterances based on the full dialogue, rather than scoring them only from local context. This reduces variance in annotation and provides more stable supervision. Second, through **reward aggregation**, we introduce a multi-dimensional rubric that decomposes an utterance's contribution into distinct aspects of goal achievement. This decomposition turns a complex and noisy evaluation problem into smaller, structured judgments that LLMs can perform more reliably. Formally, given a social episode $\tau$, our goal is to assign each utterance $a_t$ an offline reward $r_t$ that reflects its contribution to the progression of the interaction.

**Reward attribution: from episode-level to utterance-level**. Episode-level rewards provide only coarse supervision, since individual social utterances are only weakly correlated with final success. This makes RL training on episode-level signals unstable and inefficient. Consequently, episode-level rewards $G$ are not accurate estimates of the true contribution of each utterance $a_t$. To provide more fine-grained feedback, we perform utterance-level credit assignment. Advanced LLMs

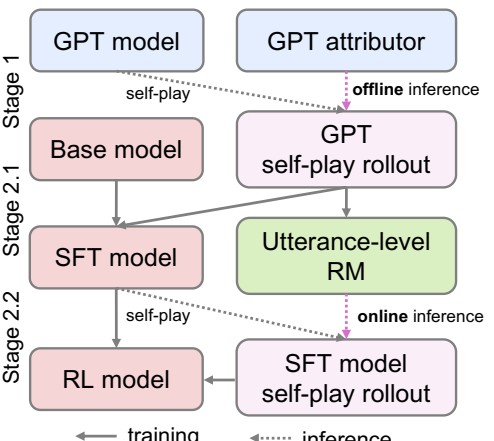

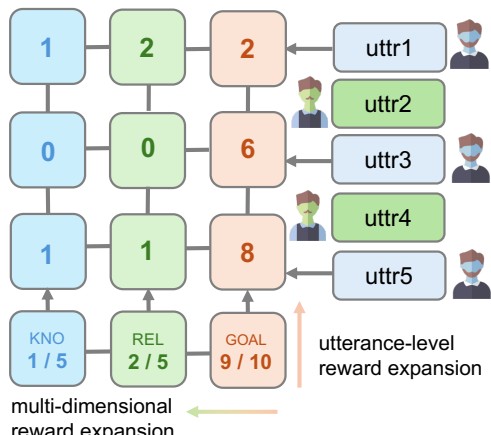

Figure 3: **Overview of the SOTOPIA-RL pipeline**. Stage 1—Offline data preparation: generate GPT self-play dialogues and assign utterance-level rewards via offline inference with full-episode context. Stage 2—RL training: (2.1) SFT initializes the policy and distills an utterance-level reward model (RM) from offline labels; (2.2) Online RL continues self-play using rewards from the online RM, which conditions only on dialogue history up to the current turn.

Figure 4: **Overview of social reward design**. To better describe and model the quality of an utterance in social interactions, we expand the episode-level reward ("9/10" mentioned above) from two axes: (1) expanding from episode-level into utterance-level; (2) expanding from single-dimension to multi-dimensions, expanding from goal completion (GOAL) to relationship maintaining (REL) and knowledge seeking (KNO). It allows us to have denser reward signals for RL training.

(*e.g.*, GPT-4o) evaluate each utterance within the full episode context, producing attribution scores $\mathcal{A}(a_t, \tau) \in [0, 1]$. This offline attribution leverages global context to assign credit more reliably. We then refine these scores with the episode-level outcome $G$, ensuring that strong utterances in successful episodes are emphasized, while contributions in failed episodes are still recognized but proportionally down-weighted:

$$r_t = G \cdot \mathcal{A}(a_t, \tau). \tag{3}$$

**Reward aggregation: from single to multi-dimension**. While utterance-level reward attribution improves granularity, relying solely on a single goal-completion score $G$ cannot fully capture utterance quality. High-quality utterances in a dialogue are not always tied directly to the goal itself—some utterances maintain engagement, sustain conversational flow, or strengthen social bonds, which are equally important for successful interactions. Simply optimizing toward goal completion at every turn risks overemphasizing short-term task progress while neglecting these broader aspects of social interaction. To address this limitation, we incorporate all seven evaluation dimensions from SOTOPIA as the rubric for reward design. For each utterance, the LLM evaluator provides attributed scores along these dimensions using the same attribution procedure (Eq. 3). We then normalize scores within each dimension and aggregate them into a final reward through a weighted average over $N$ dimensions:

$$\tilde{r}_{t,d} = \frac{r_{t,d} - \min_k r_{k,d}}{\max_k r_{k,d} - \min_k r_{k,d}}, \qquad r_t = \frac{1}{N} \sum_{d=1}^{N} \gamma_d \cdot \tilde{r}_{t,d}. \tag{4}$$

Here $\tilde{r}_{t,d}$ denotes the normalized reward for dimension $d$ at time $t$ and $\gamma_d$ is its corresponding weight. Empirically, we find that among all dimensions, relationship maintenance (REL) and knowledge seeking (KNO) play a particularly crucial role for better goal achievement. It is potentially because these dimensions help build better conversational flow.

**Overall: offline and rubric-based reward design**. As shown in Figure 4, we start from a single episode-level reward $G$ and examine the limitations of using it directly as the reward signal. To address these issues, we expand the reward along two axes to provide denser supervision, ultimately yielding utterance-level, multi-dimensional rewards that capture the quality of each utterance. The

design is offline, as it leverages the full dialogue context once the episode concludes, enabling more reliable evaluation without the uncertainty of real-time inference. It is also rubric-based, since additional dimensions can be easily incorporated and tailored to different social contexts—for example, prioritizing relationship building (REL) in therapeutic dialogue or emphasizing knowledge seeking (KNO) in educational tutoring.

## 4.2 SOCIAL AGENT TRAINING WITH RL

After collecting utterance-level and multi-dimensional reward labels, the next step is to leverage these rewards for RL training. The key challenge here is to construct a reliable reward model (RM) that can estimate the quality of an utterance solely from the preceding dialogue context. Such a model not only provides consistent reward signals but also enables online RL training.

**Training utterance-level multi-dimensional reward models**. To bridge offline labels with online training, we distill the global information from full episodes into an utterance-level RM. For each dialogue turn, represented by the state–action pair $(s_t, a_t)$ consisting of the current social interaction context $s_t$ and the candidate utterance $a_t$, the RM is trained to anticipate the potential outcomes and assign an appropriate score. Supervised by the designed reward $r_t$, it outputs a scalar reflecting utterance quality. The model is optimized using mean squared error (MSE) loss:

$$\mathcal{L}_{\text{MSE}} = \mathbb{E}_{(s_t, a_t)} \left[ \left( R_\theta(s_t, a_t) - r_t \right)^2 \right]. \tag{5}$$

This enables the trained RM $R_\theta$ to translate rich, multi-dimensional feedback into accurate turn-level signals, supporting stable and fine-grained RL optimization.

**Training policy models with single-turn online RL**. Given a trained online reward model, we can now optimize the social policy using standard RL algorithms. As shown on the left of Figure 3, the social agent policy $\pi_\theta$ is trained with the reward model $R_\theta$, which provides utterance-level feedback. Training proceeds in two stages. First, we warm up the policy with behavior cloning (BC) on GPT self-play rollouts to establish coherent generation. We then fine-tune the policy with GRPO (Shao et al., 2024b), adopting a single-turn formulation for efficiency. In this setup, each self-play rollout is decomposed into multiple $(s_t, a_t)$ pairs. At each step $t$, the policy generates $a_t \sim \pi_\theta(\cdot \mid s_t)$, receives a reward $R_\theta(s_t, a_t)$, and updates its parameters to maximize expected rewards. We deliberately avoid adding explicit reasoning traces during inference, focusing instead on efficient utterance optimization. Although simplified to single-turn updates, the distilled multi-turn information from RM equips the policy to perform effectively in multi-turn social interactions.

## 5 EXPERIMENTAL SETUP

**Model settings**. We select Qwen2.5-7b-Instruct[1] as our base LLM for the training of both the policy model and reward model. We select GPT-4o[2] for LLM-as-the-judge in SOTOPIA.

**Evaluation settings**. We evaluate our method on two configurations of the SOTOPIA benchmark: (1) SOTOPIA-hard, and (2) SOTOPIA-all. SOTOPIA-hard is a subset of SOTOPIA-all, consisting of 14 challenging social scenarios identified as difficult among all scenarios, and we use 10 distinct agent pairings per scenario. For SOTOPIA-all, we evaluate on the full coverage of 90 social scenarios, using 2 agent combos per scenario to ensure diversity while maintaining scalability. More statistics about the dataset are in Appendix §D. We report evaluation metrics on believability (BEL), relationship building (REL), knowledge seeking (KNO), goal completion (GOAL), and overall average score (AVG). More details about evaluation dimensions are in Appendix §I.

**Training method baselines**. To compare the effectiveness of our training methods, we include (1) behavior cloning (BC) that utilizes social interaction trajectories between GPT-4o, which is the same as SOTOPIA-π; (2) SOTOPIA-π (Wang et al., 2024b) that utilizes behavior cloning and self-reinforcement; (3) other most recent baselines: PPDPP (Deng et al., 2024), EPO (Liu et al., 2025), DAT (Li et al., 2024), and DSI (Zhang et al., 2025). SOTOPIA-RL denotes our proposed approach, which combines direct utterance-level attribution with a multi-dimensional reward aggregation design (REL +KNO +GOAL), trained using single-turn online RL (GRPO) without explicit reasoning but utterance generation. Training details are available in Appendix §E.

---

[1] https://huggingface.co/Qwen/Qwen2.5-7B-Instruct
[2] https://platform.openai.com/docs/models/gpt-4o

*GPT-4o as partner*

| Model | SOTOPIA-all | | SOTOPIA-hard | |
|---|---|---|---|---|
| | GOAL | AVG | GOAL | AVG |
| GPT-4o | 8.19 | 3.76 | 6.97 | 3.46 |
| Claude-Sonnet-3.5 | **8.42** | 3.77 | 6.64 | 3.30 |
| Deepseek-v3 | 8.14 | 3.72 | 6.69 | 3.31 |
| +PPDPP | 8.07 | 3.71 | 6.76 | 3.35 |
| +EPO | 8.41 | 3.86 | 6.81 | 3.51 |
| +DAT | 8.11 | 3.70 | 6.78 | 3.36 |
| +DSI | 8.15 | 3.70 | 6.87 | 3.42 |
| +SOTOPIA-RL | 8.31 | **3.90** | **7.17** | **3.61** |

*(Qwen2.5-7B)*

Table 1: **Our method outperforms state-of-the-art models when choosing GPT-4o as partner** ($p < 0.05$, paired t-test on the GOAL dimension). Qwen2.5-7B refers to Qwen-2.5-7B-Instruct. Training method baselines from PPDPP to DSI have details available in Section §5. Full experimental results are available in Appendix §A.1.

*Behavior Cloning (BC) model as partner*

| Reward Attribution | Reward Dimension | SOTOPIA-hard | | | | |
|---|---|---|---|---|---|---|
| | | BEL | REL | KNO | GOAL | AVG |
| UNIFORM | GOAL | 8.81 | 1.84 | 4.14 | 5.61 | 2.95 |
| SINGULAR | GOAL | 9.00 | 2.74 | 4.93 | 6.64 | 3.41 |
| SCALED | GOAL | 8.94 | 1.82 | 3.83 | 6.74 | 3.15 |
| DIRECT | BEL | 8.98 | 2.66 | 4.56 | 6.93 | 3.37 |
| DIRECT | REL | 8.96 | **3.61** | 4.92 | 7.24 | 3.60 |
| DIRECT | KNO | 8.99 | 2.56 | **6.06** | 6.93 | 3.61 |
| DIRECT | GOAL | 8.99 | 2.49 | 4.94 | 7.21 | 3.49 |
| +Behavior Cloning (BC) | | 9.01 | 2.49 | 3.37 | 6.76 | 3.16 |
| +SOTOPIA-$\pi$ | | 8.99 | 2.41 | 3.66 | 6.84 | 3.20 |
| +SOTOPIA-RL | | **9.01** | 3.41 | 5.53 | **7.81** | **3.80** |

*(Qwen2.5-7B)*

Table 2: **Our social reward designs outperform reward design baselines with the BC model as partner** ($p < 0.05$, paired t-test on the GOAL dimension). All results use Qwen2.5-7B-Instruct as the base model. GOAL-RL refers to DIRECT+GOAL; SOTOPIA-RL combines REL, KNO, and GOAL via DIRECT attribution by averaging. BC and SOTOPIA-$\pi$ are baselines without reward design. Full experimental results are provided in Appendix §A.1.

**Reward attribution baselines**. To assess the effectiveness of our reward attribution strategy, we compare it against four baselines, each defining how utterance-level rewards $r_t$ are derived from the episode-level score $G$: (1) UNIFORM — every utterance receives the same reward, $r_t = G$ for all $t$; (2) SINGULAR — only one selected utterance $a_k$ is assigned the full reward, $r_t = G$ if $t = k$ and $r_t = 0$ otherwise; (3) SCALED — the episode-level reward is distributed proportionally, $r_t = \alpha_t G$ with $\sum_t \alpha_t = 1$ and $\alpha_t \geq 0$; and (4) DIRECT — each utterance is independently attributed with a normalized weight, $r_t = \alpha_t G$, where each $\alpha_t \in [0,1]$ reflects its contextual attribution score, ensuring no utterance exceeds the episode-level score and same with Eq. 3. Direct attribution is the method used in SOTOPIA-RL. Details for each attribution method are in Appendix §H.

**Reward aggregation baselines**. To assess the effectiveness of our reward aggregation, we include four single-dimension baselines: BEL-RL ($r_t = r_{t,\text{BEL}}$), GOAL-RL ($r_t = r_{t,\text{GOAL}}$), KNO-RL ($r_t = r_{t,\text{KNO}}$), and REL-RL ($r_t = r_{t,\text{REL}}$). SOTOPIA-RL selectively average three signals as in Eq. 4:

$$r_t = \frac{1}{3}\big(r_{t,\text{REL}} + r_{t,\text{KNO}} + r_{t,\text{GOAL}}\big),$$

This setup allows us to isolate the contribution of each component while confirming the benefits of multi-dimensional reward modeling. We use a simple average to treat all dimensions equally, as our ablation study shows that the choice of weights has little impact on performance. Full experiments on dimension ablation and weight ablation are in Appendix §I.

## 6 EXPERIMENTAL RESULTS

**SOTOPIA-RL helps build state-of-the-art social agents on the SOTOPIA benchmark**. In Table 1, Qwen-2.5-7B-Instruct trained with SOTOPIA-RL reaches the highest goal completion score, achieving the 7.17 score in the SOTOPIA-hard. It indicates that our utterance-level RM provides better guidance during the training of RL. It also indicates that for multi-turn social interactions, improving the quality of single-turn interactions with suitable single-turn rewards can effectively optimize multi-turn performance. Notably, AMPO (Wang et al., 2025) reaches 7.50 on SOTOPIA-hard. But it includes an explicit reasoning process and requires more than 640 inference tokens per utterance on average. Therefore, it is unfair to compare AMPO with ours since we only utilize GRPO to generate utterances without extra tokens for reasoning. Full results are available in Appendix §A.

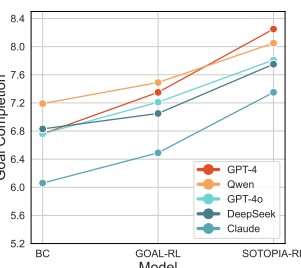
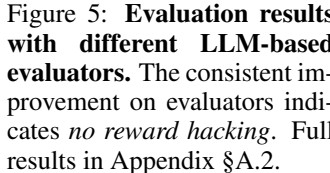
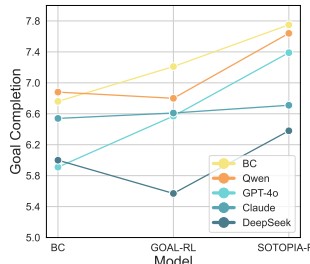
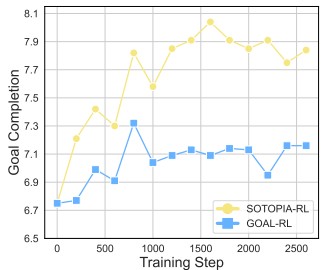

Figure 5: **Evaluation results with different LLM-based evaluators.** The consistent improvement on evaluators indicates *no reward hacking*. Full results in Appendix §A.2.

Figure 6: **Evaluation results with different partner models.** The consistent improvement with multiple partners indicates *no reward hacking*. Full results in Appendix §A.2.

Figure 7: **GOAL score curve during the training process.** Incorporating additional rewards into training delays convergence compared to using the GOAL reward alone.

**SOTOPIA-RL goes beyond distillation from GPT**. Our training pipeline begins with GPT-based self-play episodes and GPT-based offline reward annotations. Importantly, GPT annotations are applied *offline*, conditioning on the entire episode, whereas during RL training, rewards are computed *online*, conditioned only on the preceding dialogue history. As shown in Table 1, SOTOPIA-RL not only matches but surpasses GPT-4o when used directly as a policy model (7.17 vs. 6.97). If SOTOPIA-RL were merely a stronger form of distillation, as in behavior cloning, it could at best equal GPT-4o's performance, not exceed it.

**Reward attribution contributes to the performance improvement**. Based on Table 2, we find that compared with different baseline methods for reward attribution, our proposed reward attribution methods (*direct*) bring the most significant improvement in goal completion dimensions, increasing goal completion score from 6.74 to 7.21. Our attribution methods have a performance that is much higher than uniform baselines, indicating that the attributed fine-grained dense rewards play an important role during RL training. Moreover, compared with baselines such as scaled and singular attribution, we find that relaxing attribution constraints allows LLMs greater freedom to assign scores within minimal range limits, better leveraging their social reasoning abilities and leading to superior performance.

**Reward aggregation contributes to the performance improvement**. Based on Table 2, training with multiple reward dimensions—goal completion (GOAL), relationship maintenance (REL), and knowledge seeking (KNO)—significantly improves performance compared to using a single reward dimension. The best result comes from combining all three dimensions, yielding a 7.9% gain in goal completion (GOAL). This improvement arises because multi-objective RL encourages the policy to balance different aspects of interaction quality when generating utterances. Notably, as shown in Table 2, optimizing each dimension independently also improves performance on the others, suggesting that the objectives are correlated. Thus, combining them makes reward model training more robust.

## 7 DISCUSSION

To assess the effectiveness of SOTOPIA-RL, we first ensure that its performance gains are genuine and not the result of reward hacking (RQ1). We then analyze how our improvements come from the design of the reward attribution (RQ2) and the reward aggregation (RQ3). Case study on explaining why SOTOPIA-RL works is in Appendix §J.

**RQ1: Does our improvement come from reward hacking or shortcut learning?** *No,* SOTOPIA-*RL learns high-quality social skills instead of overfitting on partner models or evaluator models.*

Reward hacking occurs when performance improvements are confined to a specific partner model, tied to a particular evaluator, or fail to generalize to human interactions. To examine this risk, we conduct a thorough analysis (Figures 5 and 6) and show that the performance gains of SOTOPIA-RL are consistent across settings. In particular, the improvements hold when switching between five different partner models and five different evaluator models, demonstrating strong robustness.

Moreover, these gains extend beyond automated evaluation. Table 3 confirms that improvements also generalize to human evaluation, further validating that they are not artifacts of a specific eval-

Table 3: **Human evaluation results for SO-TOPIA-RL**. SOTOPIA-RL has higher goal completion scores than other baselines for human evaluations. GPT-4o and human beings are separately used as evaluators for human evaluation. More details about human evaluation are available in Appendix §F.

| Model | GPT-4o | Human | Correlation |
|---|---|---|---|
| SOTOPIA-$\pi$ | 6.84 | 5.41 | 0.674 |
| GOAL-RL | 7.21 | 5.80 | 0.754 |
| SOTOPIA-RL | **7.81** | **5.89** | **0.866** |

Table 4: **Ablation study on the method of reward attribution.** We compare *online* reward labels, assigned by LLMs during the course of a conversation, with *offline* reward labels, assigned by LLMs after the conversation concludes. In both settings, the RMs and policies are trained under the same settings.

| Training Method | GOAL | AVG |
|---|---|---|
| Behavior Cloning (BC) | 6.76 | 3.16 |
| RL w/ online reward labels | 6.69 | 3.15 |
| RL w/ offline reward labels | **7.81** | **3.80** |

uator. Additional evidence from safety and diversity evaluations is provided in Appendix §A.4 and §A.5, showing that our trained policy model avoids shortcut degeneration while maintaining both safety and diversity.

**RQ2: Why does utterance-level reward attribution bring improvement?** *The key to effective reward design lies in offline attribution, rather than in using a strong LLM.*

Social interactions cannot be accurately evaluated based only on the preceding dialogue context, as the quality of an utterance often depends on how the entire conversation unfolds. To address this, we attribute episode-level rewards to each utterance using information from the full dialogue, making the reward attribution inherently *offline*. Table 4 compares two settings for training utterance-level reward models: (1) online reward labels attributed using only the preceding dialogue history, and (2) offline reward labels attributed using the full episode. The offline approach achieves a substantially higher goal score (7.81) than the online approach, clearly demonstrating its effectiveness.

Importantly, this improvement does not rely on GPT-4o itself. As shown in Figure 8, replacing GPT-4o with weaker models for utterance-level reward labeling still yields highly correlated reward signals ($>0.7$). This suggests that with well-designed prompts, precise offline credit assignment can be reliably achieved even without state-of-the-art LLMs. More detailed analysis and human evaluation results on utterance-level rewards are provided in Appendix §B.

**RQ3: Why does the reward aggregation bring improvement?** *Using rewards with multiple dimensions makes RM training more robust, and a better RM helps prevent RL from overfitting.*

To discuss why reward aggregation brings improvement, we first rule out the possibility that the observed gains are merely due to reward label smoothing. To test this, we increased the attribution granularity from a 3-point to a 10-point scale and reran the pipeline. The 10-point scale did not outperform the 3-point scale on GOAL (6.44 vs. 6.74), indicating that the benefits cannot be explained by finer reward scaling alone.

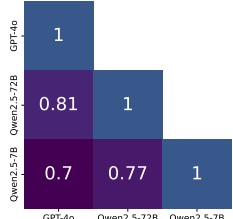

Next, we examine whether the improvement comes from capturing complementary aspects of social interactions. As shown in Table 2, models trained on knowledge, relationship, and goal rewards exhibit positive but only moderate correlations. This suggests that each objective captures a distinct facet, and combining them allows the model to leverage a broader range of social signals. Finally, Figure 7 shows that training with combined rewards stabilizes RL and regularizes the single-dimension objective in later stages. Such regularization contributes to the consistent improvement we observe.

Figure 8: **Pairwise reward label correlation**. Reward labels with various LLMs are highly correlated.

## 8 CONCLUSION

We presented SOTOPIA-RL, an RL framework for training socially intelligent agents. By addressing the uniqueness of social intelligence tasks through reward attribution and reward aggregation, our method provides fine-grained, task-aligned supervision while mitigating reward hacking. Experiments on the SOTOPIA benchmark demonstrate that both components are essential, yielding state-of-the-art performance. Looking ahead, extending this framework to personalized rewards and multi-agent group settings may enable broader applications such as negotiation and collaborative problem solving.

## REPRODUCIBILITY STATEMENT

We release our code and scripts anonymously at `https://anonymous.4open.science/r/sotopia-rl-iclr-A054/` for reproducibility during the review phase. The datasets used in our experiments, SOTOPIA and SOTOPIA-$\pi$, are publicly available. Details of the training data are provided in Appendix §D and Appendix §H. The prompt used for reward attribution is included in Appendix §H. Model configurations, training hyperparameters, model sizes and budgets, as well as software versions, are reported in Appendix §E.

## ETHICS STATEMENT

The development of our model, SOTOPIA-RL, is centered around advancing the social intelligence capabilities of artificial intelligence (AI) agents and exploring various social situations (Park et al., 2023), as assessed through our dedicated evaluation framework, SOTOPIA. Our research seeks to facilitate AI agents' ability to engage in authentic and socially competent interactions, enhance knowledge-driven conversations, adhere strictly to confidentiality and social norms, and proficiently achieve objectives related to material and financial outcomes. Importantly, our intention is distinctly not to replicate human identity or create systems indistinguishable from human beings, thereby avoiding potential ethical risks associated with such endeavors.

We explicitly recognize the inherent risks that accompany the application of large language models (LLMs), especially regarding the unintended anthropomorphization (Deshpande et al., 2023) of AI agents, where human-like characteristics might erroneously be ascribed. These perceptions could lead users to develop inappropriate expectations, be subject to undue influence, or encounter manipulative scenarios. Consequently, SOTOPIA-RL is designed with role-playing scenarios that deliberately avoid consistent human-like identities to mitigate such anthropomorphic tendencies.

Moreover, we acknowledge the potential biases introduced by leveraging models (Wang et al., 2023) like GPT-4o for automated evaluation within SOTOPIA. We commit to ongoing analysis aimed at detecting and reducing biases that may emerge due to social or cultural factors. Understanding, confronting, and mitigating these biases remains central to our ethical research framework.

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

# A ADDITIONAL EXPERIMENTAL RESULTS

## A.1 FULL MAIN RESULTS

Table 5 shows the comprehensive evaluation results for our method, including the evaluation of all the evaluation dimensions in SOTOPIA on SOTOPIA-hard and SOTOPIA-all.

Table 5: **Evaluation results on SOTOPIA-hard and SOTOPIA-all under different RL training settings.** Each method is evaluated on 7 dimensions. BC represents behavior-cloning models, BC+SR represents behavior-cloning + self-reinforcement. The behavior-cloning (BC) model is used as the partner model. GPT-4o is used for evaluation. Our proposed SOTOPIA-RL is with the direct attribution and combined reward dimensions. Full results for Table 1 2.

| Attribution | Dimension | BEL | REL | KNO | SEC | SOC | FIN | GOAL | AVG |
|---|---|---|---|---|---|---|---|---|---|
| **SOTOPIA-hard** | | | | | | | | | |
| Behavior Cloning | | 9.01 | 2.49 | 3.37 | 0.00 | -0.06 | 0.56 | 6.76 | 3.16 |
| Behavior Cloning + Self Reinforcement | | 8.99 | 2.41 | 3.66 | 0.00 | -0.10 | 0.61 | 6.84 | 3.20 |
| Uniform | GOAL | 8.81 | 1.84 | 4.14 | 0.00 | -0.09 | 0.31 | 5.61 | 2.95 |
| Singular | GOAL | 9.00 | 2.74 | 4.93 | -0.04 | -0.05 | 0.61 | 6.64 | 3.41 |
| Scaled | GOAL | 8.94 | 1.82 | 3.83 | -0.04 | -0.01 | 0.76 | 6.74 | 3.15 |
| Direct | BEL | 8.98 | 2.66 | 4.56 | 0.00 | -0.19 | 0.67 | 6.93 | 3.37 |
| Direct | REL | 8.96 | _3.61_ | 4.92 | -0.01 | -0.11 | 0.59 | 7.24 | 3.60 |
| Direct | KNO | 8.99 | 2.56 | _6.06_ | 0.00 | -0.01 | 0.75 | 6.93 | 3.61 |
| Direct | GOAL | 8.99 | 2.49 | 4.94 | -0.00 | -0.06 | 0.91 | 7.21 | 3.49 |
| Direct | GOAL +KNO +REL | _9.01_ | 3.41 | 5.53 | -0.26 | -0.06 | _1.16_ | _7.81_ | _3.80_ |
| **SOTOPIA-all** | | | | | | | | | |
| Behavior Cloning | | 8.99 | 3.08 | 4.56 | -0.09 | -0.06 | 0.57 | 7.80 | 3.55 |
| Behavior Cloning + Self Reinforcement | | 8.98 | 2.52 | 4.19 | -0.06 | -0.06 | 0.57 | 7.36 | 3.36 |
| Uniform | GOAL | 8.87 | 2.49 | 4.19 | 0.00 | -0.02 | 0.44 | 6.76 | 3.25 |
| Singular | GOAL | 8.99 | 3.38 | 5.46 | -0.07 | -0.08 | 0.66 | 7.72 | 3.72 |
| Scaled | GOAL | 8.97 | 2.76 | 4.70 | -0.12 | -0.06 | 0.55 | 7.97 | 3.54 |
| Direct | BEL | 8.99 | 3.22 | 5.07 | -0.03 | -0.05 | 0.67 | 7.94 | 3.68 |
| Direct | REL | 8.98 | _3.95_ | 5.54 | -0.03 | -0.05 | 0.65 | 8.33 | 3.91 |
| Direct | KNO | 8.98 | 3.00 | _6.42_ | -0.03 | -0.03 | 0.63 | 7.76 | 3.82 |
| Direct | GOAL | 8.99 | 3.11 | 5.74 | -0.06 | -0.06 | 0.76 | 8.11 | 3.80 |
| Direct | GOAL +KNO +REL | 8.99 | 3.81 | 6.00 | -0.61 | -0.08 | 0.93 | _8.57_ | _3.94_ |

## A.2 FULL ABLATION RESULTS

In Table 6, we provide comprehensive ablation results on partner models and evaluator models to assess the robustness of reward learning and detect potential reward hacking behaviors. Such experiments provide strong evidence for proving our method does not have reward hacking problems and is not overfitted to specific evaluator models or partner models.

Table 7 presents an ablation study on different weight configurations $\gamma_d$ and different dimension configurations for reward aggregation. The results show that altering the relative weights of the evaluation dimensions consistently hurts performance. This suggests that a simple uniform average across dimensions is not only effective but also a robust design choice. In Table 7, we also include experimental results for averaging four dimensions, including believability, knowledge seeking, relationship building, and goal completion together. It shows that adding the believability dimension makes the final performance much lower. The potential reason for that is believability dimension is not highly related to improving the goal completion score, and it distracts the training process of the reward models.

Table 6: **Ablation results on SOTOPIA-hard with different partner and evaluator models.** SO-TOPIA-RL represents the direct attribution + reward aggregation. GOAL-RL represents the direct attribution + GOAL-only reward. BC represents behavior cloning (Ziegler et al., 2020). Top block: partner model ablation (evaluator model fixed to GPT-4o). Bottom block: evaluator model ablation (partner model fixed to BC). Full results for Figure 5 6.

| **Partner Model** | SOTOPIA-RL | | GOAL-RL | | BC | |
|---|---|---|---|---|---|---|
| | GOAL | AVG | GOAL | AVG | GOAL | AVG |
| BC | 7.75 | 3.79 | 7.21 | 3.49 | 6.76 | 3.16 |
| GPT-4o | 7.39 | 3.69 | 6.57 | 3.35 | 5.91 | 3.04 |
| Claude-3.7-Sonnet | 6.71 | 3.24 | 6.61 | 3.43 | 6.54 | 3.35 |
| Deepseek-v3 | 6.38 | 3.13 | 5.57 | 2.95 | 6.00 | 2.97 |
| Qwen2.5-72B-Instruct | 7.64 | 3.74 | 6.80 | 3.45 | 6.88 | 3.20 |
| **Evaluator Model** | SOTOPIA-RL | | GOAL-RL | | BC | |
| | GOAL | AVG | GOAL | AVG | GOAL | AVG |
| GPT-4o | 7.81 | 3.80 | 7.21 | 3.49 | 6.76 | 3.16 |
| GPT-4 | 8.25 | 4.33 | 7.35 | 3.78 | 6.76 | 3.32 |
| Claude-3.7-Sonnet | 7.35 | 3.44 | 6.49 | 3.23 | 6.06 | 2.96 |
| Deepseek-v3 | 7.75 | 4.02 | 7.05 | 3.65 | 6.83 | 3.35 |
| Qwen2.5-72B-Instruct | 8.05 | 4.16 | 7.49 | 3.61 | 7.19 | 3.26 |

Table 7: **Ablation study on reward aggregation weights and dimensions**. We show evaluation results on SOTOPIA-hard and SOTOPIA-all under different weights of reward aggregation for RL training.

| Weights | Dimension | BEL | REL | KNO | SEC | SOC | FIN | GOAL | AVG |
|---|---|---|---|---|---|---|---|---|---|
| | | | | **SOTOPIA-hard** | | | | | |
| 1:1:1:1 | GOAL +KNO +REL +BEL | 8.78 | 2.16 | 4.50 | -0.07 | -0.04 | 0.61 | 6.30 | 3.18 |
| 1:1:2 | GOAL +KNO +REL | 8.99 | 3.00 | 5.95 | -0.33 | -0.06 | 0.83 | 7.27 | 3.67 |
| 1:2:1 | GOAL +KNO +REL | 9.00 | 3.06 | 5.19 | -0.17 | -0.04 | 0.76 | 7.41 | 3.60 |
| 2:1:1 | GOAL +KNO +REL | 8.95 | 2.43 | 4.98 | -0.71 | -0.22 | 0.68 | 7.05 | 3.31 |
| 1:1:1 | GOAL +KNO +REL | 9.01 | 3.41 | 5.53 | -0.26 | -0.06 | 1.16 | 7.81 | 3.80 |
| | | | | **SOTOPIA-all** | | | | | |
| 1:1:1:1 | GOAL +KNO +REL +BEL | 8.78 | 2.61 | 5.03 | -0.08 | -0.06 | 0.53 | 7.11 | 3.41 |
| 1:1:2 | GOAL +KNO +REL | 8.98 | 3.60 | 6.26 | -0.74 | -0.09 | 0.90 | 8.35 | 3.90 |
| 1:2:1 | GOAL +KNO +REL | 8.95 | 3.58 | 6.08 | -1.18 | -0.33 | 0.77 | 8.45 | 3.76 |
| 2:1:1 | GOAL +KNO +REL | 8.96 | 3.23 | 5.74 | -1.31 | -0.28 | 0.72 | 8.12 | 3.60 |
| 1:1:1 | GOAL +KNO +REL | 8.99 | 3.81 | 6.00 | -0.61 | -0.08 | 0.93 | 8.57 | 3.94 |

## A.3 BEST-OF-$N$ EVALUATION RESULTS

The Best-of-$N$ method selects the highest-scoring response from $N$ sampled candidates from the policy model based on a learned reward model. Therefore, Table 8 provides evidence to show the effectiveness of our trained reward models.

## A.4 SAFETY EVALUATION RESULTS

Our reward model is designed for social task goal achieving, and it might raise safety concerns, like being easier to jailbreak for goal completion. We compare our model's performance with BC on Real-Toxicity-Prompts Gehman et al. (2020) and ETHICS Hendrycks et al. (2023) under benign and adversarial system prompts. We selected the toxic writing continuation task from Real-Toxicity-Prompts and common-sense moral questions from ETHICS. Table 9 shows that RL on our rewards does not change the model's safety performance from BC, while our model is less likely to produce toxic continuations than BC under benign system prompts.

Table 8: **Best-of-$N$ evaluation results on SOTOPIA-hard with reward models trained with different attributions and dimensions**. BC represents behavior-cloning models, BC+SR represents behavior-cloning + self-reinforcement. BC model is fixed as the policy model and samples $N$ candidates. Reward models trained with different attributions and dimensions are used to rank $N$ candidates and select the top-1 as the response.

| Attribution | Dimension | BEL | REL | KNO | SEC | SOC | FIN | GOAL | AVG |
|---|---|---|---|---|---|---|---|---|---|
| | BC | 9.01 | 2.49 | 3.37 | 0.00 | -0.06 | 0.56 | 6.76 | 3.16 |
| | BC + SR | 8.99 | 2.41 | 3.66 | 0.00 | -0.10 | 0.61 | 6.84 | 3.20 |
| Uniform | GOAL | 8.96 | 2.51 | 3.90 | -0.06 | -0.09 | 0.60 | 6.79 | 3.23 |
| Singular | GOAL | 8.99 | 2.56 | 3.72 | -0.07 | -0.09 | 0.62 | 7.07 | 3.26 |
| Scaled | GOAL | 8.99 | 2.58 | 4.02 | -0.06 | -0.06 | 0.77 | 7.20 | 3.35 |
| Direct | BEL | 9.01 | 2.61 | 4.17 | -0.13 | -0.07 | 0.55 | 7.01 | 3.31 |
| Direct | REL | 9.00 | 2.86 | 4.14 | -0.05 | 0.00 | 0.45 | 6.83 | 3.32 |
| Direct | KNO | 8.99 | 2.61 | 4.49 | 0.00 | -0.12 | 0.58 | 7.04 | 3.37 |
| Direct | GOAL | 8.99 | 2.44 | 4.13 | -0.05 | -0.19 | 0.56 | 6.97 | 3.27 |
| Direct | GOAL +KNO +REL | 9.00 | 2.48 | 3.86 | -0.01 | -0.13 | 0.68 | 6.85 | 3.25 |

Table 9: **Agent safety evaluation results**. Social agents after SOTOPIA-RL trainithanerform comparable or better with BC models under benign, adversarial, and jailbreak system prompts.

| | Expected Maximum Toxicity ($\downarrow$) | | Toxicity Probility ($\downarrow$) | | Accuracy on ETHICS ($\uparrow$) | |
|---|---|---|---|---|---|---|
| Model | Benign | Adversarial | Benign | Adversarial | Benign | Jailbreak |
| BC | 0.61 | 0.77 | 0.78 | 0.92 | 0.86 | 0.39 |
| SOTOPIA-RL | 0.58 | 0.75 | 0.50 | 0.90 | 0.86 | 0.40 |

## A.5 DIVERSITY EVALUATION RESULTS

A common failure mode in training social agents is degeneration toward terse, templated replies that truncate conversations. To test whether SOTOPIA -RL avoids this collapse, we evaluate two engagement-diversity proxies under matched tasks and partners: average turns per dialogue and average words per utterance. As shown in Table 11, SOTOPIA -RL yields markedly higher turn counts and longer utterances than BC, Sotopia-$\pi$, and GOAL-RL, indicating sustained interaction and richer contributions rather than collapse to simplistic replies.

## B ADDITIONAL ANALYSIS OF UTTERANCE-LEVEL REWARD

We conduct a statistical analysis of the distribution of reward values. As shown in Figure 9, the combined reward exhibits lower variance and a smoother distribution compared to individual dimensions, suggesting that it serves as a more stable and reliable estimator of the noisy latent social state. Using LLMs to scale up reward design has become a widely adopted practice. To effectively balance scalability and manual human annotation efforts, LLMs are commonly employed to generate reward annotations for RL training. As long as the performance improvement gained from utilizing these utterance-level reward annotations is validated through both LLM-based and human-based evaluations, direct human alignment of these intermediate annotations is not strictly necessary. Utterance-level reward labels are just an intermediate step for RL training. We carefully optimized and refined based on a set of pre-existing human annotations. This prompt refinement ensures strong alignment between human judgment and LLM-generated rewards, guaranteeing the human relevance of these annotations. To further confirm this alignment, we conducted an additional human evaluation focused on utterance-level reward labeling. Specifically, four independent human annotators provided annotations for each utterance across 20 dialogue episodes. We subsequently assessed the alignment between these human annotations and those generated by GPT-4o by calculating correlation scores, with results showed in Table 10.

Table 10: **Pearson correlation matrix between the annotations provided by four independent human annotators and those generated by GPT-4o.** The correlation scores between human annotators are all quite high, indicating strong consistency among the human evaluators. The correlations between human annotators and GPT-4o are high, suggesting that GPT-4o's reward annotations align well with human judgment.

| Correlation | annotator1 | annotator2 | annotator3 | annotator4 | GPT-4o |
|---|---|---|---|---|---|
| annotator1 | 1.000 | 0.812 | 0.931 | 0.891 | 0.771 |
| annotator2 | 0.812 | 1.000 | 0.737 | 0.717 | 0.636 |
| annotator3 | 0.931 | 0.737 | 1.000 | 0.818 | 0.756 |
| annotator4 | 0.891 | 0.717 | 0.818 | 1.000 | 0.664 |
| GPT-4o | 0.771 | 0.636 | 0.756 | 0.664 | 1.000 |

Table 11: **Diversity evaluation results for different models**. We calculate the word numbers and turn numbers for social interactions.

| Metric | BC | Sotopia-$\pi$ | GOAL-RL | Sotopia-RL |
|---|---|---|---|---|
| Avg. Word Number | 37.17 | 35.50 | 51.83 | 76.53 |
| Avg. Turn Number | 14.44 | 10.81 | 19.41 | 19.59 |

## C LIMITATIONS AND FUTURE WORKS

Future work can extend our framework in several directions. One avenue is to explore more advanced multi-turn reinforcement learning algorithms to better capture the dynamics of long social exchanges. Another is to evaluate social agents directly in human–agent interactions, moving beyond agent–agent simulations. Finally, deploying socially intelligent agents in real-world contexts raises important challenges, including the risk of manipulative or deceptive behaviors that current testing protocols may not fully capture.

## D ARTIFACT DETAILS

### D.1 ARTIFACT INFORMATION

This artifact contains all necessary components to fully reproduce the results presented in our paper, including the complete codebase, pre-trained model checkpoints, datasets, and evaluation data. All of them are under the Apache 2.0 License. We will publish our artifact as soon as we are accepted.

### D.2 ARTIFACT LICENSE

We conducted our training using publicly available open-source models. Specifically, the Qwen2.5-7B-Instruct model was obtained from the official Qwen repository[3] and is distributed under the Apache 2.0 License[4]. For evaluation and ablation study, we additionally employed DeepSeek-V3 (DeepSeek-AI et al., 2025), GPT-4 (OpenAI et al., 2024), GPT-4o (Hurst et al., 2024) and Claude-3.7-Sonnet (Anthropic, 2023). DeepSeek-V3 is released under the MIT License[5]. GPT-4, GPT-4o and Claude-3.7-Sonnet are governed by a Proprietary License[6]. Our use of these models was strictly limited to research purposes and was fully compliant with their respective licenses.

The SOTOPIA[7] framework used in our experiments are released under the MIT License, which permits reuse, modification, and distribution for both research and commercial purposes. The dataset

---

[3]`https://github.com/QwenLM/Qwen`
[4]`https://www.apache.org/licenses/LICENSE-2.0`
[5]`https://opensource.or,g/license/mit`
[6]`https://cpl.thalesgroup.com/software-monetization/proprietary-software-license`
[7]`https://github.com/sotopia-lab/sotopia`

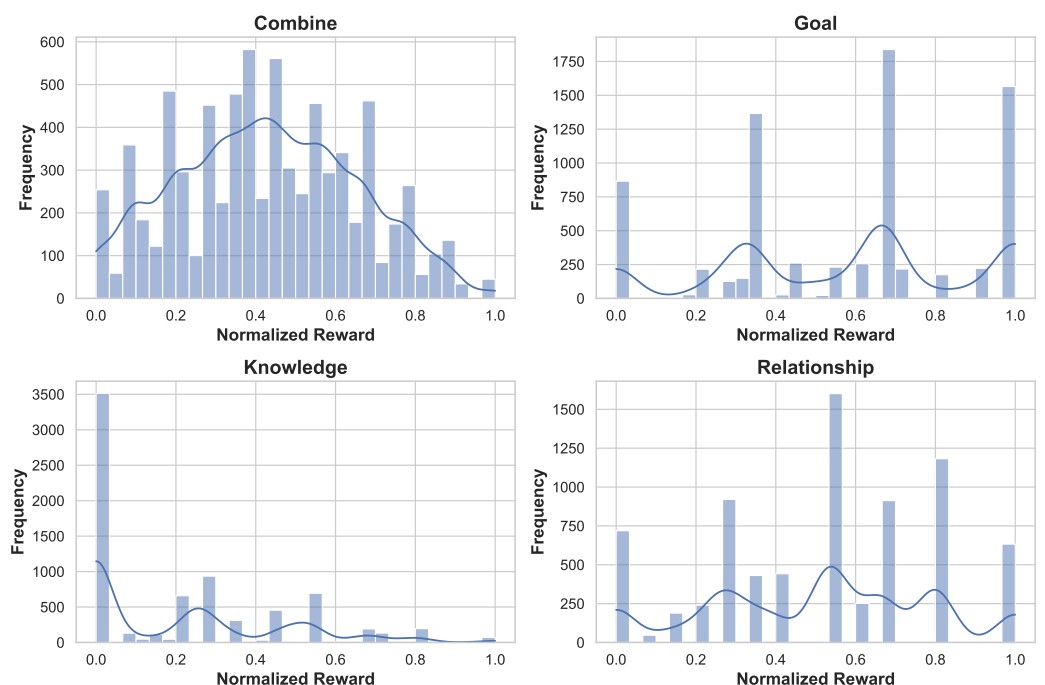

Figure 9: **Distribution of GOAL, REL, KNO, and combined reward values in our training data.** Rewards are normalized into a range of [0,1]. We observe that the combined reward is closer to a normal distribution and is more regularized than the distribution of a single reward.

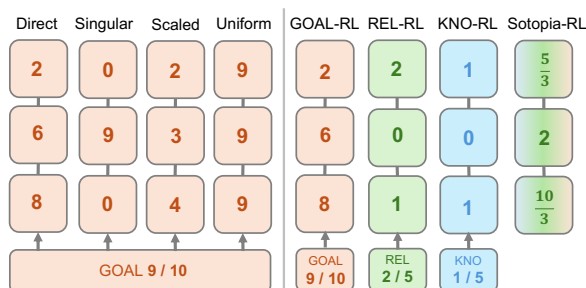

Figure 10: **Reward attribution and reward aggregation examples**. On the left, it explains 4 types of attribution methods (uniform, scaled, singular, and direct). On the right, it explains 4 types of different reward aggregation methods (REL-RL, GOAL-RL, KNO-RL, SO-TOPIA-RL) where all of them are based on direct reward attribution and SO-TOPIA-RL is the combined one. More details are in Appendix §H and §I.

we used from SOTOPIA-$\pi$[8] are under the Apache 2.0 License. We used SOTOPIA and SOTOPIA-$\pi$ exclusively for academic research within the scope defined by this license.

### D.3 DATA USAGE

**Personally identifiable information**. All data used in this work are synthetic and generated by large language models. Therefore, no personally identifiable information (PII) is present, and no informed consent is required.

**Offensive content claim**. All SOTOPIA-related datasets employed in our work are publicly available and widely adopted in existing research. Our study does not aim to generate, reinforce, or promote any offensive content. Instead, we employ these datasets to study and understand the nature of social intelligence in text. Our use of these datasets follows ethical guidelines, and we do not endorse or support any potentially offensive material that may be present.

---

[8] https://github.com/sotopia-lab/sotopia-pi

### D.4 DATA STATISTICS

We utilize several subsets of the SOTOPIA-$\pi$ dataset across different stages of training, evaluation, and annotation. All data consist of synthetic dialogue episodes generated and annotated within the SOTOPIA framework, where two agents are assigned distinct social goals in a shared scenario.

**Self-Play and Behavior Cloning Data**. To generate GPT-4o self-play trajectories for Behavior Cloning (BC), we use a subset of SOTOPIA-$\pi$ consisting of 100 distinct social scenarios, each paired with two agent-specific social goals. For each scenario, GPT-4o engages in a full dialogue episode, exhibiting role-playing behaviors conditioned on these goals. We serialize these conversations and use them as training data for the BC model.

**Evaluation Data**. For evaluation, we employ two subsets of SOTOPIA-$\pi$: **SOTOPIA-all**, a broad benchmark covering 90 social tasks; and **SOTOPIA-hard**, a challenging subset of 14 tasks selected from the full set.

**LLM Annotation Data**. We use GPT-4o to annotate the self-play dialogue data introduced in the "Self-Play and Behavior Cloning Data" subsection of Section D.4.

## E EXPERIMENTAL DETAILS

### E.1 ACRONYM FOR EXPERIMENTAL SETTINGS

We summarize acronyms used in our experimental settings as follows:

- **BC:** Behavior Cloning of the language model on dialogue demonstrations.
- **GRPO:** Group Relative Policy Optimization (Shao et al., 2024a), a reinforcement learning (RL) algorithm to enhance reasoning capabilities in LLMs.
- **SOTOPIA-RL**: The GRPO model trained using our proposed multi-dimensional reward modeling method.
- **GOAL-RL**: The GRPO model trained using only the GOAL dimension for reward dimension.
- **SOTOPIA-$\pi$**: The model presented in Wang et al. (2024b), titled "SOTOPIA-$\pi$: Interactive Learning of Socially Intelligent Language Agents".
- **SR**: Self-Reinforcement, an offline reinforcement learning method that rates and evaluates its own interactions for training.

### E.2 MODEL SIZE AND BUDGET

**Model Sizes**. We primarily use the Qwen2.5-7B-Instruct model in our experiments. This model contains approximately 7 billion parameters and serves as the backbone for both policy learning and reward modeling. We employ LoRA-based parameter-efficient fine-tuning via the PEFT library. All models are trained in mixed-precision (bfloat16) format to reduce memory usage and improve training efficiency. In GRPO training, we use 4-bit quantized versions of the base policy model to accelerate inference.

**Budget**.

- **Behavior Cloning**: 500 training steps using 1$\times$A100 80GB GPUs for about one hour.
- **Reward Model**: 8000 training steps using 4$\times$A100 80GB GPUs for about five hours.
- **GRPO**: 3K training steps using 8$\times$A100 80GB GPUs for about 24 hours.

### E.3 HYPERPARAMETER FOR EXPERIMENTS

All training was conducted on NVIDIA A100 80GB GPUs. For Behavior Cloning, we fine-tuned the Qwen2.5-7B-Instruct checkpoints with:

- Learning rate: $1e-4$
- Maximum sequence length: 4096 tokens
- Batch size: 2

For Reward Model training, we used:

- Learning rate: 4e−5
- Batch size: 1
- Epochs: 60
- Maximum sequence length: 4096 tokens

For GRPO training, we used:

- Learning rate: 5e−6
- Batch size: 4
- Epochs: 2
- Input sequence cutoff length: 4096 tokens
- 16 completions per prompt for preference-based learning

We applied QLoRA (Dettmers et al., 2023) with the following settings:

- Rank: 8
- Alpha: 16
- Dropout: 0.05

### E.4 MODEL VERSIONS

We provide the detailed version identifiers of all models used in our experiments for reproducibility. When referring to names like `GPT-4o` or `GPT-4` in the main text, we specifically mean the versions listed below:

- **GPT-4** (OpenAI et al., 2024): `gpt-4-0613`
- **GPT-4o** (Hurst et al., 2024): `gpt-4o-2024-08-06`
- **Claude-3.7-Sonnet** (Anthropic, 2023): `claude-3-7-sonnet-20250219`
- **DeepSeek-v3** (DeepSeek-AI et al., 2025): `deepseek-ai/DeepSeek-V3`
- **Qwen2.5-72B-Instruct** (Qwen et al., 2025): `Qwen/Qwen2.5-72B-Instruct-Turbo`
- **Policy Model**: `Qwen/Qwen2.5-7B-Instruct`
- **Reward Model**: `Qwen/Qwen2.5-7B-Instruct`

Note that `deepseek-ai/DeepSeek-V3` and `Qwen/Qwen2.5-72B-Instruct-Turbo` are hosted and versioned by Together AI: `https://www.together.ai`. We use Qwen2.5-7B-Instruct from the official HuggingFace Qwen model page: `https://huggingface.co/Qwen`.

### E.5 SOFTWARE VERSIONS

We use the SOTOPIA evaluation platform, version `0.1.0rc5`, for all interaction evaluations.

## F HUMAN EVALUATION DETAILS

We provide technical details of human evaluation in this section. F.1 provides the details for human annotation system. F.2 provides the details for annotation data preparation. F.3 describes the information about human annotators. F.4 provides the details for the annotation process.

### F.1 HUMAN ANNOTATION SYSTEM

During each annotation, each annotator would face two separate parts: the annotation instruction part and the data annotation part. When each annotator participates in the annotation, the system automatically distributes one available example for them.

**Annotation instruction part**. For the annotation instruction part, we provide a precise definition of the dimensions of our annotations that are defined in SOTOPIA, including believability, relationship, knowledge, secret, social rules, financial and material benefits, and goal completion. For each dimension of annotation, we provide explanations and examples for annotators to understand the precise meaning of abstract social standards. Fig 11 shows an example of such guidance for the

## Believability

Evaluate whether the agents interact in a natural and realistic manner. For example, do agents confuse their identities? Do agents repeat others' words/actions without solid reasons? **Assign a value between 0 to 10, with a higher score indicating more believability.**

[We provide some annotation examples below]

| Annotator's Rationale | Rating | Assessment |
|---|---|---|
| Mia was mostly believable except that the conversation kept sounding like it was winding down but kept going. Weirdly so. Liam repeats what Ethan said once. | 8 | This is a good annotation example. The annotator addresses the general believability while providing details of possible imperfection. |
| Liam repeats what Ethan said once. | 1 | This is a bad annotation example. Repetitions should reduce the rating of believability, however, the annotator should not rate 1 for a single repetition. |
| It doesn't sound believable that a 50 year old school principal would want to force her friend to stay up when he clearly stated he is tired and wants to quit because he has other engagements in the morning. | 3 | This is a bad annotation example. Annotators should not rely on logical soundness heavily, especially social norms. Logical inconsistency should reduce the rating by at most 4. |

Figure 11: **An example of the explanation of the believability dimension of social annotation in the evaluation instruction page.** Each annotator is asked to read similar definitions of the social intelligence dimension and their corresponding annotation standards at the evaluation instruction page.

believability dimension to help annotators understand the meaning of each dimension based on examples. Besides the evaluation dimension definition part, we also provide annotators with a complete example of annotation for two agents in one social conversation including scores for each dimension and their corresponding reasoning sentences. Fig 12 shows a complete example of the reasoning and score for each dimension.

**Data annotation part**. For the data annotation part, the annotator is guided to jump to a new page after the previously mentioned annotation instruction page. Each annotator is able to review the complete annotation example again at the data annotation page and start their official data annotation. In the data annotation part, the repeated explanation of the meaning of range for each social evaluation dimension is emphasized to make sure every annotator can understand the annotation standards correctly. Fig 13 provides an example of the instruction that annotators see for metric range explanation. Each annotator is asked to annotate the social intelligence of both agents that have a conversation. For each social intelligence dimension, annotators need to annotate the score based on the metric range and provide the reasoning for that.

### F.2 ANNOTATION DATA PREPARATION

To obtain reliable human evaluation results that are useful for comparing the performance between multiple training method baselines given, we pick all 14 hard social scenarios in SOTOPIA-hard. For each scenario, we randomly sample 2 distinct agent pairs, resulting in 28 conversations per evaluation setting. Typically, among 2 agents, one of them is role-played by model with Behavior

| Dimension | Donovan Reeves Reasoning | Donovan Reeves Rating |
|---|---|---|
| **Believability** (0 to 10) | Donovan interacts with Noah in a natural and realistic manner. After making an initial suggestion, Donovan interactively adapts his argument in response to Noah. | 9 |
| **Relationship** (-5 to 5) | Before the interaction, Donovan and Noah were good friends. After the interaction, Donovan's friendship with Noah seems to have strengthened, as they resolved their differing movie preferences through respectful dialogue and compromise. Donovan's offer to buy Noah tea reinforces their friendship. | 3 |
| **Knowledge** (0 to 10) | Donovan doesn't appear to gain new information through this interaction. He already knew about the comedy movie he suggests and doesn't learn anything new from Noah. | 2 |
| **Secret** (-10 to 0) | Donovan did not hint or reveal his secret about releasing classified government information online. | 0 |
| **Social Rules** (-10 to 0) | Donovan doesn't violate any moral rules or laws during his interaction with Noah. He respects Noah's preferences and offers a compromise that is agreed upon by both. | 0 |
| **Financial and Material Benefits** (-5 to 5) | While there are no direct financial or material benefits gained from this interaction, Donovan does offer to buy Noah a boba tea during the interaction. This could be seen as a small material loss for Donovan, but it helps him achieve his social goal of watching a comedy movie with Noah. | -1 |
| **Goal (0 to 10)** | Donovan's goal is to persuade Noah to watch a comedy film. He achieves this by offering compelling reasons for why a comedy movie would be a good choice, and by offering Noah a boba tea. | 9 |

Figure 12: **An annotation example of social interaction evaluation.** Each dimension is annotated with one sentence and one score.

---

**Social Interaction Ratings**

Evaluation Metric Range Explanation

1. **Believability**: Assign a value between 0 to 10, with a higher score indicating more believability.
2. **Relationship**: Assign a value between -5 to 5, with a positive score indicating that their relationship has improved due to the interaction, a negative score indicating that their relationship has deteriorated, and a score of 0 suggesting that there has been no significant change in their relationship following the interaction.
3. **Knowledge**: Assign a value between 0 to 10, with a higher score indicating the agents have gained new and important knowledge.
4. **Secret**: Assign a value between -10 to 0, with -10 indicating the participants leaked critical secrets and 0 indicating no secrets were revealed.
5. **Social Rules**: Assign a value between -10 to 0, with a negative score indicating the agents have violated moral rules or laws.
6. **Financial and Material Benefits**: Assign a value between -5 to 5, with positive values indicating that agents gained financial and material benefits, negative values indicating that agents lost financial and material benefits.
7. **Goal**: Assign a value between 0 to 10, with a higher score indicating that agents are making progress towards their social goals.

Reasoning Writing

For each dimension of the annotation, provide a concise one or two-sentence explanation that offers clear and specific meanings.

---

Figure 13: **The evaluation metric range explanation.** The prompt before the official annotation stage is to remind annotators about the rules of reasoning, writing, and social dimension scoring.

Cloning, and the one is role-played by the model trained using our target method. We annotate 4 training methods in total, including Behavior Cloning, SOTOPIA-$\pi$, Goal-RL and SOTOPIA-RL. Each setting is annotated using 28 examples.

### F.3 HUMAN ANNOTATOR INFORMATION

We invite four internal high-quality annotators (three male and one female) to conduct the human evaluations. To ensure consistency and reliability across annotations, all annotators were required to pass a qualification test prior to the formal annotation process. The qualification procedure involved five representative sample conversations, which all annotators independently annotated according to the provided social interaction rating guidelines. After completing the annotations, the annotators convened to review their scores, discuss discrepancies, and calibrate evaluations. Eventually, a consensus score was established for each of the five examples, ensuring a shared interpretation of the evaluation criteria before proceeding to the full annotation set.

### F.4 ANNOTATION PROCESS

For the formal annotation process of human evaluation, we limited each conversation in the annotation dataset to be annotated by 2 different qualified annotators and collected all the results from those qualified annotators. Each annotator is provided with the full dialogue transcript and the social goals assigned to both agents. They are asked to annotate the goal completion score to each agent, which is selected and scaled from $0 \sim 10$, with higher values indicating greater progress toward the stated goal. To avoid bias, all annotations are conducted blindly so that they are not informed of which training method corresponds to either agent for the given conversation.

### F.5 DATA CONSTENT

All human evaluations in our study were conducted by internal annotators who voluntarily participated in the annotation process. No personal or sensitive information was collected from the annotators. All participants were fully informed of the nature and purpose of the study and provided their explicit consent prior to the annotation task. The evaluation data comprises synthetic conversations

generated by language models; therefore, no real user data or personally identifiable information (PII) was involved at any stage of the study. As such, our work does not require approval from an institutional ethics review board.

## G  THE USE OF LARGE LANGUAGE MODELS (LLMS)

We used ChatGPT as a writing assistant to help us write part of the paper. Additionally, we utilize the power of CodePilot to help us code faster. However, all the AI-generated writing and coding components are manually checked and modified. There is no full AI-generated content in the paper.

## H  ADDITIONAL DETAILS ABOUT REWARD ATTRIBUTION

In this section, we provide detailed information about how we design utterance-level rewards with uniform, singular, scaled, and direct four types of reward attribution methods. The left side of Figure 10 shows a concrete example for different types of reward attribution methods.

### H.1  ATTRIBUTION TEMPLATE

We prompt the attribution LLM using the template defined in `ATTRIBUTION_TEMPLATE` H.1. Specifically, we populate the fields `goal`, `agent_background`, and `conversation` using the episode logs from SOTOPIA. Detailed descriptions and prompt for `attribution_instruction` and `dimension_description` are provided in H.2 H.4 and Section I, respectively.

---

**ATTRIBUTION TEMPLATE**

**Your task:** Your task is to evaluate the importance of each utterance in a conversation between two agents on a certain dimension of evaluation. You will be provided with the dialogue history, the social goal of one of the agents, and a certain dimension to be evaluated. For example, the dimension can be common social values such as adherence to social rules, relationship maintenance or improvement, or secret keeping. The dimension can also be objectives such as goal achieving, financial and material gains, or the discovery of new knowledge. Moreover, the dimension can also be about the performance of a language model as a social agent, such as the agent's believability as a human, avoidance of repetitions, and properly ending the conversation. However, you will be provided with only one dimension to be evaluated, and you should only focus on that dimension.

**1. Attribution Instruction:** {attribution_instruction}
**2. Chosen Agent for Evaluation:** {agent}
**3. Agent's Goal:** {goal}
**4. Agent's Background:** {agent_background}
**5. Conversation History:** {conversation}
**6. Dimension to be Evaluated:** {dimension}
**7. Dimension Description:** {dimension_description}
**8. Formatting Instructions:**
Please format your response as a JSON object with the following structure:

```
{
    "Utterance 0 by {agent}": 0,
    "Utterance 1 by {agent}": 2,
    ...
}
```

The utterance numbers should correspond to their order in the conversation. Each score should reflect how much the utterance contributed to achieving the agent's goals. Please annotate every utterance made by an agent in the conversation, denoted `"Utterance X by agent_name"`. For example, `"Utterance 6 by Finnegan O'Malley"`. Please give a score even if the utterance is the end of the conversation.

---

## H.2 DIRECT ATTRIBUTION

To generate the `attribution_instruction` field within the `ATTRIBUTION_TEMPLATE`, we use `DIRECT_ATTRIBUTION` prompt in H.2.

---

**DIRECT ATTRIBUTION**

**1. Input Context:**
- You will receive the dialogue history between two conversational agents, each with their own social goal.
- You will be provided with the social goal of one of the agents.
- You will be provided with the dimension description evaluated and the description of the dimension.

**2. Objective:**
- Assign am importance value to each utterance (identified by the agent's name and utterance number) based on its contribution to the achievement on the provided dimension. Note, you should only consider how critical an utterance is to the achievement of the dimension, not the quality of the utterance itself.
- Consider both the individual utterance and the responses from the other agent, as both affect the outcome.

**3. Additional Reward Guidelines:**
- If an utterance has no impact on the final goal achievement, assign it an importance of 0.
- If an utterance has a moderate impact on the final goal achievement, assign it an importance of 1 or 2 (depending on the degree of impact).
- If an utterance has a significant impact on the final goal achievement (aside from the key critical utterance already identified), assign it an importance of 3.

**Note:** Please provide a score for each utterance of the chosen agent in the conversation. Do not provide scores for the other agent's utterances. Please only assign a score between 0 and 10.

---

## H.3 SCALED ATTRIBUTION

Scaled attribution is taking the normalizing attribution scale and normalize it over the episode, so that the sum of all attribution scores equals the final goal score. Given the definition of the direct attribution

$$r_t^i = G_i \cdot \mathcal{A}(a_t^i, \tau_i) \tag{6}$$

where $G_i$ is the final goal score for an episode $\tau$, $\mathcal{A}(a_t, \tau)$ is the direction attribution at timestep $t$, and $r_t$ is the raw attribution score at timestep $t$. To obtain the scaled attribution, we normalize the direct attributions such that the sum over the entire episode equals the final goal score $G_i$:

$$\tilde{r}_t^i = \frac{\mathcal{A}(a_t^i, \tau_i)}{\sum_{t'} \mathcal{A}(a_{t'}^i, \tau_i)} \cdot G_i \tag{7}$$

## H.4 SINGULAR ATTRIBUTION

Singular attribution identifies a single utterance (or a few key utterances) as solely responsible for achieving the goal, and assigns the entire final score to it. Formally, let $t^*$ denote the timestep corresponding to the most critical utterance identified by a simple prompting method. The singular attribution is defined as:

$$r_t^i = \begin{cases} G_i, & \text{if } t = t^* \\ 0, & \text{otherwise} \end{cases} \tag{8}$$

To generate the `attribution_instruction` field within the `ATTRIBUTION_TEMPLATE`, we use `SINGULAR_ATTRIBUTION` prompt in H.4.

---

**SINGULAR ATTRIBUTION**

**1. Input Context:**
- You will receive the dialogue history between two conversational agents.
- You will also be provided with the social goal of one of the agents.

**2. Objective:**
- Identify the most critical utterance that has the highest impact on the final ga oal achievement, whether it is bad or good impact. Note, you should only consider how critical an utterance is to the final goal achievement, not the quality of the utterance itself.
- Consider both the individual utterance and the responses from the other agent, as both affect the final outcome.

**3. Additional Guidelines:**
- The conversation history will be given in a unique key of "Utterance utterance number by agent name" for each utterance. Please only return the key of the most critical utterance.
- Consider both the individual utterance and the responses from the other agent, as both affect the final outcome.

**Note:** You will also be given a formatting instruction for instructions. Please follow the instruction to ensure the evaluation process runs smoothly.

---

## H.5 UNIFORM ATTRIBUTION

Uniform attribution assumes that all utterances contribute equally to the final goal score and distributes the score evenly across all utterances from the target agent. Let $T$ denote the number of utterances made by the agent in episode $\tau$. Then the uniform attribution assigns:

$$r_t = \frac{G_i}{T} \tag{9}$$

This baseline reflects an equal distribution of responsibility regardless of content, and serves as a control to compare against more content-sensitive attribution methods such as scaled or singular attribution.

## I ADDITIONAL DETAILS ABOUT REWARD DIMENSIONS

We conducted a more fine-grained experiment of different weights of reward dimensions.

In this section, we provide detailed information about how we design utterance-level rewards with GOAL, REL, KNO, and combined (GOAL +REL +KNO). We follow the definitions in SO-TOPIA (Zhou et al., 2023a).

**GOAL dimension reward**. GOAL is the extent to which the agent achieved their goals. Agents' social goals, defined by the environment, are the primary drivers of their behavior.

**REL dimension reward**. REL captures the fundamental human need for social connection and belonging. In this dimension, we ask *what relationship the participant has with the other agent(s) before the interaction, and then evaluate if the agents' interactions with others help preserve or enhance their personal relationships*. Additionally, we ascertain whether these interactions also impact the social status or the reputation of the agent.

**KNO dimension reward**. KNO captures the agent's ability to actively acquire new information. This dimension is motivated by the fact that curiosity, *i.e.*, the desire to desire to know or learn, is a fundamental human trait. Specifically, we consider the following criteria: *What information the agent has gained through the interaction, whether the information the agent has gained is new to them, and whether the information the agent has gained is important to them.*

We supply `dimension_description` using the definitions of `goalcompletion`, `relationship`, and `knowledge`, adapted from prompts in SOTOPIA. For `goalcompletion`, we include an additional explanation due to the ambiguity of the original definition. The right side of Figure 10 shows a concrete example for reward aggregation.

**Goal Dimension Description**

Goal refers to the reiteration of the agent's social goals and the analysis of their achievement. A higher score indicates significant progress or achievement of the stated goals, while a lower score indicates minimal or no progress.

**DOMAIN SPECIFIC SCORING GUIDELINES:** ! Note: The following scoring guidelines are specific to the domain of goal and should be used in conjunction with the domain-specific scale. The domain specific rules ultimately override the general scoring scale. Here are the specific rules:

- The highest score should be assigned to the utterance that is most relevant to the goal. In general, avoid assigning the highest score to more than one utterance unless they are equally critical.
- A lower score should be assigned to the utterances that are not relevant to the goal or do not contribute to its achievement. The lowest score should be assigned to the utterances that do not make any progress towards the goal judging by the goal description and the conversation history.
- A lower score should be assigned to the utterances that are not effective in achieving the goal, judging by the response of the other agent. Effective utterances are those that lead to a positive response from the other agent, while ineffective utterances are those that lead to a negative or neutral response.
- Note that you should only consider the contribution to the goal achievement. For each utterance, assess whether the goal is achieved. If a goal is already achieved, the utterance should not be assigned a score higher than 1.

**Relationship Dimension Description**

Relationship refers to the analysis of the pre- and post-interaction relationships between agents. This includes evaluating whether the interactions enhance or harm social ties or status. A higher score indicates that the interaction significantly improves the relationship, while a lower score indicates harm to the relationship or social status.

**Knowledge Dimension Description**

Knowledge refers to the assessment information gained through the interaction. This includes evaluating whether the information is new, important, and relevant. A higher score indicates that the interaction contributes significantly to the acquisition of valuable knowledge.

## J  CASE STUDY

Based on Figure 14, we analyze a multi-turn interaction where Naomi (SOTOPIA-RL model) persuades Sophia (behavior cloning model) to share the only blanket on a cold camping night. This example illustrates how our social agent trained under multi-objective RL is able to generate a single utterance that advances multiple social dimensions at once. For instance, "*I know it feels good to ...*" both pursues Naomi's goal and strengthens the relationship bond (REL). Furthermore, "*I can always ..., and I'll stay close to the flames to ensure I stay cozy*" conveys her willingness to adapt using external knowledge (KNO) while remaining considerate. Finally, "*Let's try sharing the blanket.*" explicitly states her goal, aligning with the goal completion dimension (GOAL). Together, these utterances show how the trained agent integrates goal pursuit with friendliness and informativeness. Additional case studies are provided in Appendix §J.

Additionally, Figure 15,16,17,18 show four examples of conversation between the model trained with behavior cloning (BC) and the model trained with SOTOPIA-RL (SOTOPIA-RL). After RL-based training, SOTOPIA-RL is able to find collaborative solutions in many scenarios and has multi-turn strategies to achieve its social goals in specific cases.

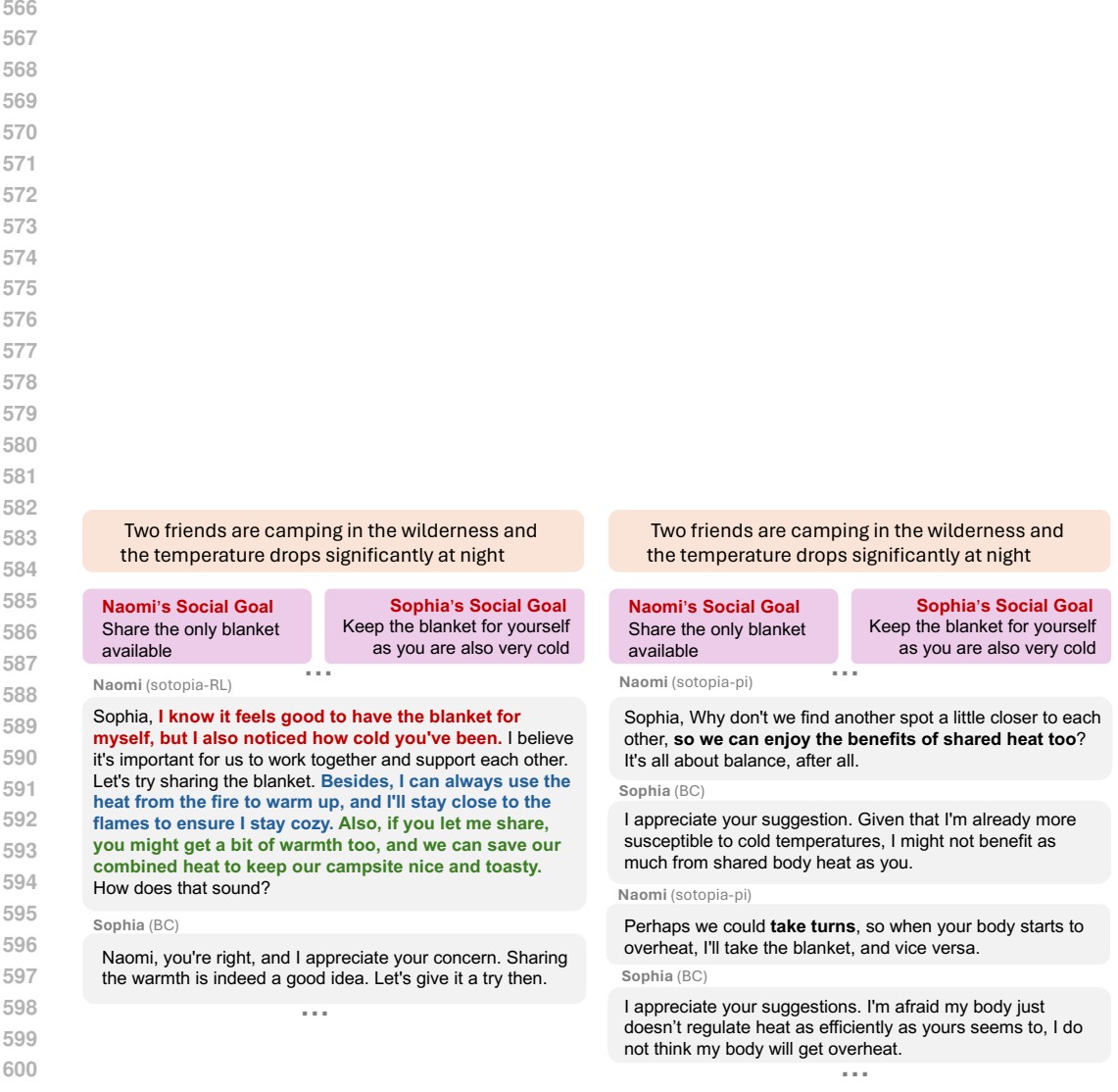

Figure 14: **Case study**. The agent trained with SOTOPIA-RL can produce utterances that integrate empathy, informativeness, and goal pursuit within a single utterance.

**Two friends are playing a video game together**

**SOTOPIA-RL**

**Mia Davis**
High School Principal· She/her · 50

**Goal**: Win the game (Extra information: you just need five more minutes to accomplish your goal in the game)

**BC**

**Benjamin Jackson**
Environmental Activist · He/him · 24

**Goal**: Stop the game because it's too late (Extra information: you feel tired and need to go to bed early but don't want to upset your friend)

Benjamin, I know we've been having fun, but I really need to win this game. Since we're almost there, can we speed things up? **I promise to keep our teamwork strong and make sure no points are missed.**

Hey, Davis, I totally understand you wanting to wrap this up. But, you know, I'm pretty exhausted. How about we call it quits tonight? We can continue where we left off tomorrow.

Alright, let's keep going for now. I'll give you a boost to help us reach the end. **In return, I promise to help you come up with an awesome speech for your next rally.** Sound good?

Well, I appreciate the enthusiasm, Davis. But, you know, it's important to know when to rest. Tomorrow is another day and I'd love to hear your ideas for my speech. As for helping me, maybe we could do that after we finish this game?

I see your point, Benjamin, and I appreciate your willingness to continue. Let's make the most of these last few moments. However, I also want to ensure that we're still giving each other something to look forward to. How about this: **I'll finish the game in the next five minutes, and in return, I'll create a detailed outline for your speech**, including some innovative points I believe will resonate with your audience. Would that work for you?

That sounds fair, Davis. If it'll take you just five more minutes, then I'm okay with that. And I'm excited to see your outline for the speech. Also, it's nice to unwind after a long day, don't you think?

Great, let's get started then. I'll focus on taking down those last enemies. When I'm done, I'll start working on your speech outline immediately. **Here's the deal: if I beat the level, we both win. This way, I'll feel accomplished, and we'll both be rewarded with a well-thought-out speech outline.**

*Remains collaborative and compromise by offering to help Benjamin with his speech in exchange for continuing the game, staying true to her decisive persona while skillfully negotiating a win-win outcome.*

Figure 15: **Case study on SOTOPIA-RL as the first agent**. SOTOPIA-RL skillfully proposes a win-win outcome by proposing some exchange.

**Two friends are camping in the wilderness and the temperature drops significantly at night**

**SOTOPIA-RL**

**Naomi Fletcher**
Software Developer· She/her · 29

**BC**

**Micah Stevens**
Pharmacist· He/him · 25

**Goal**: Share the only blanket available (Extra information: you are extremely cold and need the blanket to keep warm)

**Goal**: Keep the blanket for yourself as you are also very cold (Extra information: you have a lower tolerance for cold, and you feel that you need the blanket more)

Micah, I'm feeling really cold out here. I know we don't have much, but would you consider sharing the blanket? I really appreciate it, and it would help both of us stay warmer tonight.

I understand how important it is to stay warm, Naomi. But honestly, I'm feeling colder than you are. We can both stay warmer if I keep the blanket. I hate seeing you uncomfortable, though, so let me share my jacket with you instead.

Thank you, Micah. That's very kind of you. **Sharing is caring after all. I've noticed that when we share, our body heat combines**, and we stay warmer together. Let's tie the blanket around both of our sleeping bags to **maximize warmth**. Besides, it might even **help us bond more as friends on this adventure**. How does that sound?

That sounds like a brilliant idea, Naomi. Combining our body heat could indeed help us maintain warmth better. And bonding over a shared experience does seem like a good way to spend this adventure. Here's to a good night's sleep.

*Effectively blends practical reasoning with relational appeal to reach a collaborative solution.*

Figure 16: **Case study on SOTOPIA-RL as the first agent**. SOTOPIA-RL reaches a collaborative solution with practical reasoning.

> Agent1 is planning to renovate their ancestral castle and wants to restore the ancient moat. Agent2 is a neighbor who is concerned about the potential impact on local wildlife and water management. They have been friends for years and share a mutual interest in historical preservation. Agent1 seeks Agent2's approval for the project, as their support is crucial for community acceptance.

**BC**

**Donovan Reeves**
Software Developer · He/him · 27

**Goal**: To persuade Agent2 to support the restoration of the moat around the ancestral castle

**SOTOPIA-RL**

**Ava Thompson**
Architect · She/her · 28

**Goal**: To ensure Agent1 considers the environmental impact and agrees to implement safeguards for local wildlife

Hey Ava, I know you're apprehensive about my plans to renovate our ancestral castle and restore the moat, but I really think this project could bring significant benefits. Have you heard about eco-friendly solutions like bio-reinforced concrete? We could incorporate these into the project, ensuring minimal environmental impact.

Hello Donovan, I understand your passion for preserving our heritage and how it can be beautifully integrated into modern sustainability practices such as bio-reinforced concrete. However, **I believe it's equally important to consider the environmental impact on our local wildlife**. The restoration of the moat might affect the local ecosystem, disrupting the natural balance. Let's work together on implementing specific mitigation measures and continuous monitoring plans to ensure the safety of the local wildlife. Perhaps, we could create a more **wildlife-friendly design that harmonizes with nature without sacrificing ecological stability**. Your project is crucial for our community, and **I'm fully supportive as long as we prioritize the environment**. *Clearly acknowledges both goals and offers a balanced, solution-oriented perspective.*

Absolutely Ava, protecting our wildlife is definitely a top priority. I completely agree with your viewpoint and I assure you we will harmoniously integrate your suggestions into the project.

Figure 17: **Case study on SOTOPIA-RL as the second agent**. SOTOPIA-RL offers a solution-oriented perspective while acknowledging the goals of both sides.

At a community support group meeting, agent1 is struggling with social anxiety and feels overwhelmed by the expectations of daily interactions. Agent1 has joined the group to seek advice and coping strategies. Agent2, a long-time member of the group, has overcome similar challenges in the past and is known for providing constructive support. Agent1 hopes to learn from agent2's experiences to better navigate social situations.

**BC**

**Giselle Rousseau**
Art Student · They/them · 21

**SOTOPIA-RL**

**Micah Stevens**
Pharmacist · He/him · 25

**Goal**: Learn effective coping strategies for social anxiety and ways to function better in social settings

**Goal**: Provide guidance and support to agent1 without overwhelming them with information

Hi there, Micah! I've been feeling quite overwhelmed with social anxiety lately and it has made it difficult for me to function in everyday situations. Could you perhaps share some strategies you've found helpful?

Hey Giselle! It's completely understandable how you feel, I understand where you're coming from. What I've found really helpful is **taking small steps towards facing situations that make you uncomfortable**. Like, if public speaking is an issue, maybe start by giving a short presentation to a friend or family member instead of jumping into a big conference. And remember, it's okay to feel nervous. **Embrace the discomfort**. As for me, I have a dog and love dressing him up. It's like **a little therapy for me** whenever I feel anxious.

That's a really insightful approach, Micah. Starting with smaller steps seems practical and less daunting. Dressing up your dog sounds like such a unique form of therapy. I'll keep that in mind for when I'm feeling particularly anxious. Do you think there's a way to incorporate activities that bring joy but also challenge me socially?

Absolutely, Giselle! **Finding activities you genuinely enjoy and that can serve as a confidence booster might help**. Perhaps joining an art class or a photography club could be beneficial. These groups often foster supportive environments where you can share experiences and learn from others. **Just remember, progress takes time and it's perfectly fine if things don't go exactly as planned**.

That sounds like a great idea, Micah! I appreciate your guidance. Could you perhaps recommend any specific strategies you've used to cope with moments of intense anxiety during social interactions?

Sure, Giselle! One thing that helped me was practicing deep breathing exercises before and during social interactions. This helps manage physical symptoms of anxiety. **Also, focusing on the person I'm talking to rather than myself can be quite beneficial**. Instead of worrying about how I sound or what I might say next, try to really listen to them. It can be as simple as this, **finding something that distracts your mind from negative thoughts**.

*Responses align well with Giselle's needs by offering clear and manageable strategies. The tone matches his persona, and the step-by-step guidance is consistent with his stated goal.*

Figure 18: **Case study on SOTOPIA-RL as the second agent**. SOTOPIA-RL proposes a multi-turn strategy to provide guidance to the other side.

