# OpenReview forum: "Sotopia-RL: Reward Design for Social Intelligence"
_ICLR.cc/2026/Conference — ICLR 2026 Conference Withdrawn Submission_

### Official Review · Reviewer_N1rT · 2025-10-27

**Soundness:** 2
**Presentation:** 2
**Contribution:** 2
**Rating:** 2
**Confidence:** 4

**Summary:**

This paper proposes a complete reinforcement learning framework for social tasks, which includes the following steps: collecting trajectories, obtaining multi-dimensional rewards for each trajectory offline using GPT, allocating trajectory rewards to each round of interaction offline using GPT, weighting the sum of rewards for each round of interaction offline to obtain the total reward, training a reward model, and training a strategy model using GRPO. The article conducted experiments on the SOTOPIA benchmark.

**Strengths:**

1. Proposed a feasible social task reinforcement learning framework that can improve the performance of the model on specified tasks
2. The article uses multidimensional rewards, which enhances the robustness and density of the rewards
3. The article uses a credit assignment mechanism, which can effectively allocate the overall rewards to individuals

**Weaknesses:**

1. The paper did not confirm the lack of overlap between the dataset used for GRPO and the evaluation dataset, which may have led to unfairness in results
2. The motivation for using online reinforcement learning is unclear, and the main difficulty of the article is to train an effective reward model. However, this reward model is trained using offline data, and there may be overfitting issues when used in online reforcement learning. The reward model training epoch published in the paper is 60, but the number of steps is only 8000, which increases the possibility of overfitting
3. The paper did not disclose the direct evaluation results of the rewards predicted by the reward model on the test set, only its Best-of-N performance on the benchmark, which makes it difficult to prove that the reward model strictly follows the definition of your reward
4. The reward training data of the paper requires multiple annotations using GPT, which is costly.

**Questions:**

see the above comments.

---

> ### Author Response · Authors · 2025-11-25
>
> **[train/eval overlap]** We follow the Sotopia-$\pi$[1] experimental settings and make sure there is no scenario overlap between training and Sotopia-based evaluation. Therefore, there is no potential data leakage.
>
> **[online RL and offline reward data]** The use of online RL could give the model real-time feedback after each single-turn interaction. To ensure the agent receives reliable reward signals, we first train the reward model using offline data. This ensures that the feedback provided during the online learning phase is high-quality, stable, and accurate, allowing the agent to effectively optimize its actions even in the single-turn setup. The offline training of the RM also helps avoid the scarcity of rewards. Using the offline reward data can also reduce the need for extensive online interaction data, therefore reducing the cost.
>
> **[reward model training]** We apologize for the confusion. The training epoch 60 is the maximum epoch number we set for evaluation. The real training epoch is only around 10.
>
> **[reward model evaluation]** We understand Best-of-N as a harder and more convincing case for evaluating the results of the predicted rewards. Without using the RL, we directly sample and rank with reward models. Single-turn-level utterance reward improvement eventually leads to the improvement of overall multi-turn-level reward improvement. We believe such improvement is stronger proof of the performance of the reward model under multi-turn interaction.
>
> **[cost estimation]** We first want to emphasize that, compared with human annotation, our methods are scalable and much cheaper. We roll out 5.75k episodes for reward model training. It costs <\\$100 for episode rollout and <\\$20 for reward collection. The reward collection cost is much lower than the episode rollout. We follow the Sotopia-$\pi$ recipe for episode rollout and achieve similar costs. All baselines also conduct a similar data collection process and consider it widely accepted. We also mention in Figure 8 that using smaller LLMs can still achieve high correlation for offline reward collection.
>
> [1] Wang et al. SOTOPIA-π: Interactive Learning of Socially Intelligent Language Agents

---

### Official Review · Reviewer_sYWk · 2025-11-01

**Soundness:** 2
**Presentation:** 3
**Contribution:** 2
**Rating:** 4
**Confidence:** 4

**Summary:**

The paper proposes Sotopia-RL, a two-stage framework for training socially intelligent language agents. Stage 1 converts episode-level feedback in the Sotopia environment into utterance-level, multi-dimensional rewards via an offline LLM-based attribution scheme. Stage 2 distills these labels into a reward model and optimizes a Qwen2.5-7B policy using single-turn GRPO. Sotopia-RL is an interesting and technically competent paper that makes a clear empirical contribution to the growing literature on social intelligence for LLMs. Its core idea refines coarse episode-level social feedback into utterance-level, multi-dimensional rewards.

**Strengths:**

1. The paper tackles a meaningful and underexplored challenge, which is training agents for socially grounded interaction rather than factual or reasoning tasks.

2. The two-stage pipeline (offline LLM-based attribution and online GRPO optimization) is simple, reproducible, and builds upon recent trends in process and preference-based reward modeling.

3. Results on both Sotopia-hard and Sotopia-all benchmarks show consistent improvements, with well-designed ablations that isolate contributions from the attribution and aggregation components

**Weaknesses:**

1. While the idea of converting coarse feedback into fine-grained, multi-dimensional rewards is meaningful in social RL, it is not fundamentally new in the broader LLM-RL landscape. The design closely parallels Process Reward Modeling (PRM) and other recent works on token- or step-level credit assignment in reasoning and coding domains. The novelty here mainly lies in applying such techniques to social environments rather than introducing a new RL principle.

2. The method’s heavy reliance on GPT-4o-generated labels poses concerns. These labels may not generalize well due to evaluator bias or semantic mismatch between GPT-4o’s judgment patterns and human social preferences. They can serve as part of the guidance but should ideally be combined with other supervision sources, such as human annotations, ensemble models, or behavioral regularizers, to ensure robust alignment.

3. The single-turn GRPO formulation simplifies optimization but sidesteps temporal dependencies intrinsic to conversation. Without multi-turn or trajectory-level baselines (e.g., PPO over full dialogues), it remains unclear whether the model truly learns sequential social reasoning or merely improves isolated utterance quality.

4. The reward computation, multiplicative scaling with episode-level scores and per-episode min–max normalization, is largely heuristic, with limited theoretical or empirical justification. A comparison with additive, normalized, or expectation-based credit assignment schemes would strengthen the methodological credibility.

**Questions:**

1. Your Eq. for aggregation introduces weights γ over dimensions; claiming “little impact” without numbers weakens the design choice. (In the paper this appears as the per-dimension weighting in the multi-dimensional aggregation.)
What would convince me: A table/plot sweeping γ (e.g., REL/KNO/GOAL ratios from 0–1 on a simplex) reporting GOAL/AVG with CIs on Sotopia-hard/all, plus robustness across partners/evaluators. Also include a default = equal-weights row.

2. If the partner model co-evolves, the training distribution shifts; if frozen, you risk overfitting to a fixed partner. Clarify whether the partner is frozen or updated; if frozen, add a control where partners are alternated (family + size diversity) and report generalization; if co-evolving, describe update cadence and show stability metrics (e.g., non-collapsing rewards).

3. The single-turn formulation ignores temporal dependencies that define conversation; we need to know if gains persist with trajectory-level credit. Add a multi-turn/trajectory baseline (e.g., PPO/GRPO over full dialogues or K-step returns) under the same RM and show (i) final scores, (ii) convergence curves, and (iii) qualitative examples where temporal coherence matters (goal switches, delayed reciprocity). The paper itself frames social interaction as partially observable and later approximates an MDP for single-turn optimization, please justify with comparative results.

---

> ### Author Response · Authors · 2025-11-25
>
> **[novelty]** We discuss the connection with PRM and this in the related work (Line 122-124). Our methods are considered as one type of process reward model, but precisely designed for the ambiguity and multi-dimensionality of social interactions.
>
>
> **[single-turn RL]** We focus on single-turn RL because when we have a good enough reward model that can conduct credit assignment for each utterance correctly, we have no need to conduct multi-turn RL, but can directly utilize a single-turn RL with the credit assignment score for more robust training. Such a setting does not equivalent to multi-turn RL, but provides a simplification and allows us to focus on discussing the effect of the construction of the reward model.
>
> **[ablation on gamma]** We include the ablation study on reward aggregation weights and dimensions in Table 7 (Line 885-901). Since this is a meta-level design discussion. It is nearly impossible to run the full possibility of reward combinations, but we show 4 different experimental settings for results.
>
> **[ablation on evaluator model bias]** We conduct evaluator model bias analysis in Figure 5 (Line 378-391). Changing 5 different evaluator models proves our performance gain is not biased against GPT-4o.
>
> **[ablation on partner model bias]** The partner models remain frozen during training. We show the details ablation of different partner models in Figure 6 (Line 378-391). We already run evaluations under 5 different partner models with 4 different model families (Qwen, Deepseek, Claude, GPT) to show the generalization of our improvement.
>
>
> **[multi-turn RL baselines]** We will add a multi-turn RL baseline like Archer in the later version.

---

### Official Review · Reviewer_fSpN · 2025-11-04

**Soundness:** 2
**Presentation:** 2
**Contribution:** 2
**Rating:** 2
**Confidence:** 3

**Summary:**

The paper introduces SOTOPIA-RL, a reinforcement learning (RL) framework designed to improve social intelligence in large language models (LLMs). Unlike tasks such as math or coding, social intelligence tasks (e.g., negotiation, collaboration, persuasion) require nuanced reasoning where individual utterances do not directly correspond to outcomes and success is multi-dimensional. They introduce utterance-level credit assignment, which entails decomposing coarse, episode-level feedback into per-utterance signals. Additionally, they introduce multi-dimensional rewards to expand beyond goal completion to capture dimensions like relationship maintenance and knowledge seeking, and  capture the full richness of social interactions and reduce reward hacking.

**Strengths:**

- They show superior performance on social intelligence benchmarks, scoring 7.81 on SOTOPIA-hard and 8.57 on the SOTOPIA-all dataset
- They provide a novel reward structure of Utterance-Level Credit Assignment and multi-dimensional rewards.
- They show lack of reward hacking and robustness against overfitting

**Weaknesses:**

There are several weaknesses, that I would encourage the authors to address:
- Limited human evaluation: You only conduct a small-scale human annotation study with 4 annotators (as noted in your Appendix) and your main evaluation relies on LLM-based automatic evaluators (GPT‑4o) which may not fully capture the full range of human social judgments. The number of human annotators and where they are recruited from should be in the main text. You should also note if you had an IRB for the study.
- Evaluation scenario scope: experiments are restricted to two-agent interactions of SOTOPIA, and you do not test multi-party, long horizon capabilities
- Im not sure how helpful it is to have an algorithm just focused on the SOTOPIA framework, and not looking at social interactions in general

**Questions:**

- How does your method perform on out-of-distribution social tasks or real‐world human interactions?

**Details Of Ethics Concerns:**

I'm afraid that the results of 4 annotators is used to summarize findings, which is not sufficient, and the fact that only four are in the study is not put in the main paper but only in the Appendix. They do not note if they have IRB approval, as well as do not discuss how much compensation is given. I apologize if these details were missed in my reading.

---

> ### Author Response · Authors · 2025-11-25
>
> **[human evaluation]** We include the details of the human annotator in Appendix F.3 (Line1326-1334). We also include the Pearson correlation matrix between the annotations provided by four independent human annotators and those generated by GPT-4o in Table 10 (Line 972-983). Based on the author's guidance on ICLR submission, we include the ethics statement from Line 495-512 for human evaluation.
>
>
> **[sotopia environment]** Sotopia is considered a well-known general framework for evaluating social interactions. The environment includes diverse types of social tasks, including persuasion, collaboration, accommodation, and negotiation, in Figure 1. We consider multi-party and long-horizon capabilities as different topics for reward design, and the Sotopia environment provides a diverse enough statement.
>
> **[real-world evaluation]** We believe that the Sotopia-Eval framework we used already includes a sufficiently diverse set of test cases that cover a wide range of social interaction scenarios. The variety of tasks in the Sotopia framework has different agent roles, personalities, goals, and complexities, ensuring that our model is exposed to a broad range of social challenges. Additionally, the multi-turn nature of these interactions tests the model’s ability to adapt and plan over time, providing a strong foundation for real-world social dynamics.

---

### Official Review · Reviewer_HHAm · 2025-11-07

**Soundness:** 2
**Presentation:** 3
**Contribution:** 2
**Rating:** 4
**Confidence:** 3

**Summary:**

This paper presents Sotopia-RL, a reinforcement learning (RL) framework designed to enhance the social intelligence of large language models (LLMs). The core contribution is a novel reward design that addresses two key challenges in social tasks: the weak correlation between individual utterances and final outcomes, and the multi-dimensional nature of social interactions. The method involves (1) reward attribution, which uses a powerful LLM (GPT-4o) to assign episode-level outcomes to individual utterances using full dialogue context, and (2) reward aggregation, which combines rewards from multiple social dimensions (Goal, Relationship, Knowledge) into a single signal. The authors demonstrate that agents trained with Sotopia-RL outperform strong baselines, including the base GPT-4o model, on the Sotopia benchmark, and provide extensive ablation studies and human evaluations to validate their design choices and rule out reward hacking.

**Strengths:**

Well-Motivated and Novel Problem Formulation: The paper compellingly argues that social intelligence tasks are fundamentally different from math or coding, making standard RL reward signals ineffective. The identification of "weak correlation" and "multi-dimensionality" as core challenges is precise and well-justified.
Clear and Practical Methodology: The proposed two-stage pipeline (offline reward collection, online RL training) is clearly explained and seems practically implementable. The distinction between offline attribution (with full context) and online reward modeling (with partial context) is a crucial and well-reasoned design choice.
Rigorous and Extensive Evaluation: The empirical evaluation is a major strength. The paper goes beyond simple benchmark comparisons to include:
Comprehensive Ablations: Systematic studies on the contributions of reward attribution vs. aggregation (Tables 2, 4, 5).
Robustness Checks: Tests across different partner and evaluator models (Table 6, Figures 5-6) effectively address the critical concern of reward hacking.
Human Evaluation: Corroborating automated scores with human judgment (Table 3) significantly strengthens the validity of the claims.
Safety and Diversity Analysis: Including checks for toxicity (Table 9) and conversational diversity (Table 11) shows a commendable breadth of consideration.
Strong Empirical Results: The results are impressive. Outperforming the model (GPT-4o) that was used to generate the training data and provide reward labels is a non-trivial result that demonstrates the method is doing more than simple distillation.

**Weaknesses:**

Baseline Clarification and Comparison: The definition and implementation of some baselines (e.g., PPDPP, EPO, DAT, DSI in Table 1) are not sufficiently detailed in the main text, requiring the reader to hunt through citations. A brief summary of how these methods work and why they are relevant comparators would improve clarity. Furthermore, a comparison to simpler fine-tuning methods like Direct Preference Optimization (DPO) on the same data would have been a valuable baseline.
Statistical Reporting: While the paper notes a paired t-test was used (p<0.05 for Table 1), it does not consistently report measures of variance (e.g., standard deviation) for its key results. Presenting results as single scores (e.g., 7.17, 8.31) without confidence intervals or error bars makes it difficult to fully assess the stability and significance of the improvements, especially in the ablation tables.
Writing and Presentation Issues:
There are typographical errors and formatting inconsistencies throughout the paper (e.g., "Sotopla" vs. "Sotopia," "rel training" in Fig 4 caption, "ga oal" in Appendix H.4, "Sotopia-{}\Omega}" in references). This detracts from the paper's professionalism.
The narrative flow in Sections 6 and 7 could be tighter. The "Main Discoveries" in Section 6 feel somewhat disconnected from the more detailed analysis in Section 7. Integrating these sections could improve readability.
The claim that the method is "often without costly human annotation" is slightly misleading. It relies heavily on GPT-4o for both self-play data generation and reward annotation, which is a significant computational and financial cost, even if it's not human effort.
Potential for Overfitting to Sotopia's Rubric: The reward design is tightly coupled with the specific seven-dimensional evaluation rubric of the Sotopia environment. The generalizability of the approach to other social benchmarks or real-world interactions, where such a well-defined rubric may not exist, remains an open question.

**Questions:**

Statistical Significance: Could you provide standard deviations or confidence intervals for the primary results in Tables 1 and 2? This would help the reader understand the variability in performance across different social scenarios and agent pairings.
Cost-Benefit Analysis: Given the reliance on GPT-4o for annotation, could you provide a rough estimate of the computational cost of the offline reward collection stage? How does the performance gain of Sotopia-RL compare to a simpler, less expensive method like using the same budget for more extensive supervised fine-tuning (SFT) on the GPT-4o self-play data?
Generalization: How do you envision this reward design framework being applied to a social task or environment that does not have a pre-defined, multi-dimensional evaluation rubric like Sotopia's? Is the method dependent on this structure?
Ablation on Base Model: The policy and reward model are both based on Qwen2.5-7B. To what extent do you believe the performance improvements are dependent on the choice of the base model? Have you observed similar gains when applying the Sotopia-RL pipeline to other open-source models?

---

> ### Author Response · Authors · 2025-11-25
>
> **[baseline information]** For baseline comparison, we reference [1]. (a) PPDPP, which utilizes an RL-trained policy planner to predict predefined strategies for assisting LLM reasoning; (b) EPO, which employs an RL-trained strategy reasoning LLM to generate strategies in an open-ended action space; \(c\) DAT, which uses an RL-trained planner to predict continuous action vectors for controlling LLM outputs; (d) DSI, which enhances LLM’s social capabilities through Dynamic Strategy Injection learning. They are considered various forms of RL baselines for solving social tasks.
>
> **[SFT baseline]** The Sotopia-$\pi$ baseline we included is an improved SFT baseline with the same self-play data. It filters out multi-turn conversations with high goal-achieving scores and conducts behavior cloning and self-reinforcement with episode-level rewards. The performance in Table 2 shows the improvement.
>
> **[DPO baseline]** We did not include DPO as a baseline primarily due to the lack of appropriate preference-tuning datasets. Our focus was on comparing other RL-based methods (e.g., PPDPP, EPO) that are better suited to address the reasoning challenges in social interaction tasks, which align more closely with our experimental setup.
>
>
> **[significant testing]** Since the goal dimensions have multiple discrete values ranging from 0-10, we run once under different settings and conduct the datapoint-level paired t-test under different experimental settings.
>
> **[cost estimation]** We first want to emphasize that, compared with human annotation, our methods are scalable and much cheaper. We roll out 5.75k episodes for reward model training. It costs <\\$100 for episode rollout and <\\$20 for reward collection. The reward collection cost is much lower than the episode rollout. We follow the recipe of Sotopia-$\pi$ for episode rollout and have similar costs. All baselines also conduct a similar data collection process and consider it widely accepted. We also mention in Figure 8 that using smaller LLMs can still achieve high correlation for offline reward collection.
>
> **[generalization across tasks]** The rubrics of Sotopia are not specific to typical types of social tasks. Any two-agent social interaction task has similar metrics to goal achieving to measure its performance. Sotopia itself covers diverse types of social interaction tasks (as shown in Figure 1): persuasion, collaboration, accommodation, and negotiation. Therefore, we can easily extend our framework to other social task benchmarks.
>
> **[base model ablation]** We will use llama as the base model for training in the later version. However, we would like to emphasize that our method has been experimentally proven effective with the Qwen2.5 model in the current setup.
>
>
> **[typo errors]** For the typo of "Sotopla" vs. "Sotopia" typo, we fail to find the existing mistakes pointed out. For "rel training" typo in the Figure 4 caption, we believe we do not mention this word in the Figure 4 caption. For "ga oal" typo, we would fix in the later version of our paper.
>
> **[writing disconnection]** We believe the "main discoveries" are in the introduction section instead of Section 6. Also, we believe it is inappropriate to connect the main results and the detailed analysis.
>
>
>
> [1] Wang et al. Think on your Feet: Adaptive Thinking via Reinforcement Learning for Social Agents
>
> [2] Wang et al. SOTOPIA-π: Interactive Learning of Socially Intelligent Language Agents

---

### Note · Authors · 2026-01-06

I have read and agree with the venue's withdrawal policy on behalf of myself and my co-authors.